# MAESTRO: Open-Ended Environment Design for Multi-Agent Reinforcement Learning

**Mikayel Samvelyan**[12] **Akbir Khan**[2] **Michael Dennis**[3] **Minqi Jiang**[12] **Jack Parker-Holder**[4]
**Jakob Foerster**[4] **Roberta Raileanu**[1] **Tim Rocktäschel**[2]

[1]Meta AI  [2]University College London  [3]UC Berkeley  [4]University of Oxford

samvelyan@meta.com

## Abstract

Open-ended learning methods that automatically generate a curriculum of increasingly challenging tasks serve as a promising avenue toward generally capable reinforcement learning agents. Existing methods adapt curricula independently over either environment parameters (in single-agent settings) or co-player policies (in multi-agent settings). However, the strengths and weaknesses of co-players can manifest themselves differently depending on environmental features. It is thus crucial to consider the dependency between the environment and co-player when shaping a curriculum in multi-agent domains. In this work, we use this insight and extend Unsupervised Environment Design (UED) to multi-agent environments. We then introduce *Multi-Agent Environment Design Strategist for Open-Ended Learning* (MAESTRO), the first multi-agent UED approach for two-player zero-sum settings. MAESTRO efficiently produces adversarial, joint curricula over both environments and co-players and attains minimax-regret guarantees at Nash equilibrium. Our experiments show that MAESTRO outperforms a number of strong baselines on competitive two-player games, spanning discrete and continuous control settings.[1]

## 1 Introduction

The past few years have seen a series of remarkable achievements in producing deep reinforcement learning (RL) agents with expert (Vinyals et al., 2019; Berner et al., 2019; Wurman et al., 2022) and superhuman (Silver et al., 2016; Schrittwieser et al., 2020) performance in challenging competitive games. Central to these successes are adversarial training processes that result in curricula creating new challenges at the frontier of an agent's capabilities (Leibo et al., 2019; Yang et al., 2021). Such automatic curricula, or autocurricula, can improve the sample efficiency and generality of trained policies (Open Ended Learning Team et al., 2021), as well as induce an open-ended learning process (Balduzzi et al., 2019; Stanley et al., 2017) that continues to endlessly robustify an agent.

Autocurricula have been effective in multi-agent RL for adapting to different *co-players* in competitive games (Leibo et al., 2019; Garnelo et al., 2021; Baker et al., 2019; Bansal et al., 2018; Feng et al., 2021), where it is crucial to play against increasingly stronger opponents (Silver et al., 2018) and avoid being exploited by other agents (Vinyals et al., 2019). Here, algorithms such as self-play (Silver et al., 2018; Tesauro, 1995) and fictitious self-play (Brown, 1951; Heinrich et al., 2015) have proven especially effective. Similarly, in single-agent RL, autocurricula methods based on Unsupervised Environment Design (UED, Dennis et al., 2020) have proven effective in producing agents robust to a wide distribution of *environments* (Wang et al., 2019; 2020; Jiang et al., 2021a; Parker-Holder et al., 2022). UED seeks to adapt distributions over environments to maximise some metrics of interest. Minimax-regret UED seeks to maximise the *regret* of the learning agent, viewing this process as a game between a teacher that proposes challenging environments and a student that learns to solve them. At a Nash equilibrium of such games, the student policy provably reaches a minimax-regret policy over the set of possible environments, thereby providing a strong robustness guarantee.

However, prior works in UED focus on single-agent RL and do not address the dependency between the environment and the strategies of other agents within it. In multi-agent domains, the behaviour of other agents plays a critical role in modulating the complexity and diversity of the challenges faced by a learning agent. For example, an empty environment that has no blocks to hide behind might be most

---

[1]Videos of MAESTRO agents are available at maestro.samvelyan.com

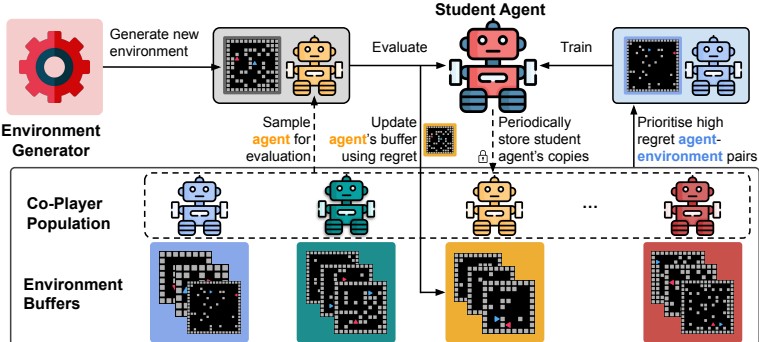

Figure 1: **A diagram of MAESTRO.** MAESTRO maintains a population of co-players, each having an individual buffer of high-regret environments. When new environments are sampled, the student's regret is calculated with respect to the corresponding co-player and added to the co-player's buffer. MAESTRO continually provides high-regret environment/co-player pairs for training the student.

challenging when playing against opponent policies that attack head-on, whereas environments that are full of winding hallways might be difficult when playing against defensive policies. Robust RL agents should be expected to interact successfully with a wide assortment of other rational agents in their environment (Yang et al., 2021; Mahajan et al., 2022). Therefore, to become widely applicable, UED must be extended to include multi-agent dynamics as part of the environment design process.

We formalise this novel problem as an *Underspecified Partially-Observable Stochastic Game* (UP-OSG), which generalises UED to multi-agent settings. We then introduce *Multi-Agent Environment Design Strategist for Open-Ended Learning* (MAESTRO), the first approach to train generally capable agents in two-player UPOSGs such that they are robust to changes in the environment and co-player policies. MAESTRO is a replay-guided approach that explicitly considers the dependence between agents and environments by jointly sampling over environment/co-player pairs using a regret-based curriculum and population learning (see Figure 1). In partially observable two-player zero-sum games, we show that at equilibrium, the MAESTRO student policy reaches a Bayes-Nash Equilibrium with respect to a regret-maximising distribution over environments. Furthermore, in fully observable settings, it attains a Nash-Equilibrium policy in every environment against every rational agent.

We assess the curricula induced by MAESTRO and a variety of strong baselines in two competitive two-player games, namely a sparse-reward grid-based LaserTag environment with discrete actions (Lanctot et al., 2017) and a dense-reward pixel-based MultiCarRacing environment with continuous actions (Schwarting et al., 2021). In both cases, MAESTRO produces more robust agents than baseline autocurriculum methods on out-of-distribution (OOD) human-designed environment instances against unseen co-players. Furthermore, we show that MAESTRO agents, trained only on randomised environments and having never seen the target task, can significantly outperform *specialist* agents trained directly on the target environment. Moreover, in analysing how the student's regret varies across environments and co-players, we find that a joint curriculum, as produced by MAESTRO, is indeed required for finding the highest regret levels, as necessitated by UED.

In summary, we make the following core contributions: (i) we provide the first formalism for multi-agent learning in underspecified environments, (ii) we introduce MAESTRO, a novel approach to jointly learn autocurricula over environment/co-player pairs, implicitly modelling their dependence, (iii) we prove MAESTRO inherits the theoretical property from the single-agent setting of implementing a minimax-regret policy at equilibrium, which corresponds to a Bayesian Nash or Nash equilibrium in certain settings, and (iv) by rigorously analysing the curriculum induced by MAESTRO and evaluating MAESTRO agents against strong baselines, we empirically demonstrate the importance of the joint curriculum over the environments and co-players.

## 2 PROBLEM STATEMENT AND PRELIMINARIES

In single-agent domains, the problem of *Unsupervised Environment Design* (UED) is cast in the framework of an underspecified POMDP (Dennis et al., 2020), which explicitly augments a standard POMDP with a set of *free parameters* controlling aspects of the environment subject to the design process. We extend this formalism to the multi-agent setting using stochastic games (Shapley, 1953).

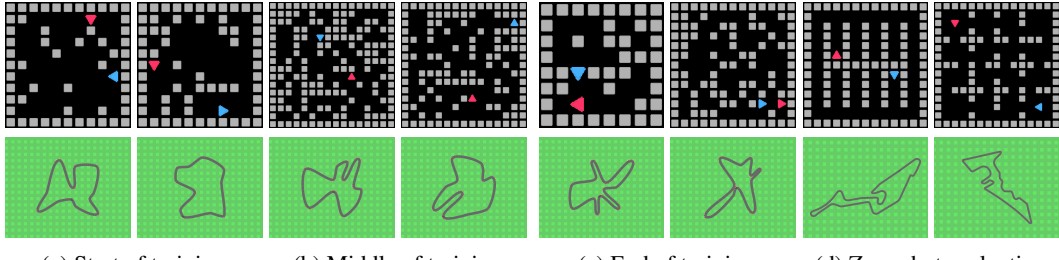

| (a) Start of training | (b) Middle of training | (c) End of training | (d) Zero-shot evaluation |

**Figure 2: Emergent complexity of autocurricula induced by MAESTRO.** Examples of partially observable environments provided to the MAESTRO student agent at the (a) start, (b) middle, and (c) end of training. Levels become more complex over time. LaserTag levels (top row) increase in wall density and active engagement between the **student** and **opponent**. MultiCarRacing tracks (bottom row) become increasingly more challenging with many sharp turns. (d) Example held-out human-designed LaserTag levels and Formula 1 benchmark tracks (Jiang et al., 2021a) used for OOD evaluation. For the full list of evaluation environments see Appendix B.

An ***Underspecified Partially Observable Stochastic Game (UPOSG)*** is given by the tuple $\mathcal{M} = \langle n, \mathcal{A}, \mathcal{O}, \Theta, S, \mathcal{T}, \mathcal{I}, \mathcal{R}, \gamma \rangle$. $\mathcal{A}, \mathcal{O}$, and $S$ denote the action, observation, and state spaces, respectively. $\Theta$ is the set of the environment's free parameters, such as possible positions of walls in a maze or levels of a game. These parameters can be distinct at every time step and are also incorporated into the transition function $\mathcal{T} : S \times \mathbf{A} \times \Theta \to \mathbf{\Delta}(S)$, where $\mathbf{A} \equiv \mathcal{A}^n$ is the joint action of all agents. Each agent draws individual observations according to the observation function $\mathcal{I} : S \times N \to \mathcal{O}$ and obtains reward according to the reward function $\mathcal{R} : S \times \mathbf{A} \times N \to \mathbb{R}$, where $N = \{1, \ldots, n\}$. The discount factor is denoted by $\gamma$. Each configuration of the parameter $\theta \in \Theta$ defines a specific instantiation of the environment $\mathcal{M}_\theta$, which is often referred to as a *level* (Jiang et al., 2021a; Parker-Holder et al., 2022). In this work, we will simply refer to a specific instance $\theta$ as an *environment* when clear from the context. We use $-i$ to notate all agents *except i*: $\boldsymbol{\pi}_{-i} = (\pi_1, ..., \pi_{i-1}, \pi_{i+1}, ..., \pi_N)$. For an agent $i$ with a stochastic policy $\pi_i : \mathcal{O} \times \mathcal{A} \to [0, 1]$, we define the value in $\mathcal{M}_\theta$ with co-players $\boldsymbol{\pi}_{-i}$ as $V^\theta(\pi_i, \boldsymbol{\pi}_{-i}) = \mathbb{E}[\sum_{t=0}^T \gamma^t r_t^i]$ where $r_t^i$ are the rewards achieved by agent $i$ when following policy $\pi_i$ in $\mathcal{M}_\theta$. The goal is to sequentially provide values for $\Theta$ and co-players $\boldsymbol{\pi}_{-i}$ to agent $i$ during training so that the policy $\pi_i$ is robust to any possible environment and co-player policies, i.e., $\pi_i = \arg\min_{\pi_i} \max_{\theta, \boldsymbol{\pi}_{-i}}(V^\theta(\pi^*, \boldsymbol{\pi}_{-i}) - V^\theta(\pi_i, \boldsymbol{\pi}_{-i}))$, where $\pi^*$ is the optimal policy on $\theta$ with co-players $\boldsymbol{\pi}_{-i}$. UPOSGs are general by nature and allow for cooperative, competitive, and mixed scenarios. Furthermore, $\Theta$ can represent different game layouts, changes in observations, and environment dynamics. When $n = 1$, UPOSGs are identical to UPOMDPs for single-agent tasks. In the remainder of this work, we concentrate on competitive settings with $n = 2$ agents.

## 2.1 UED APPROACHES IN SINGLE-AGENT RL

A curriculum over the environment parameters $\theta$ can arise from a teacher maximising a utility function $U_t(\pi, \theta)$ based on the student's policy $\pi$. The most naive form of UED is *domain randomisation* (DR, Jakobi, 1997; Sadeghi & Levine, 2017), whereby environments are sampled uniformly at random, corresponding to a constant utility $U_t^U(\pi, \theta) = C$. Recent UED approaches use *regret* as the objective for maximisation (Dennis et al., 2020; Gur et al., 2021): $U_t^R(\pi, \theta) = \max_{\pi^* \in \Pi}\{\text{REGRET}^\theta(\pi, \pi^*)\} = \max_{\pi^* \in \Pi}\{V_\theta(\pi^*) - V_\theta(\pi)\}$, where $\pi^*$ is the optimal policy on $\theta$.

Empirically, regret-based objectives produce curricula of increasing complexity that result in more robust policies. Moreover, if the learning process reaches a Nash equilibrium, the student provably follows a minimax-regret policy (Dennis et al., 2020): $\pi \in \arg\min_{\pi_A \in \Pi}\{\max_{\theta, \pi_B \in \Theta, \Pi}\{\text{REGRET}^\theta(\pi_A, \pi_B)\}\}$, where $\Pi$ and $\Theta$ are the strategy sets of the student and the teacher, respectively. While this is an appealing property, the optimal policy $\pi^*$ is unknown in practice and must be approximated instead.

*Prioritized Level Replay* (PLR, Jiang et al., 2021a;b) continually curates an *environment buffer* containing the environments generated under domain randomisation with the highest learning potential, e.g., as measured by estimated regret. PLR alternates between evaluating new environments for learning potential and performing prioritised training of the agent on the environments with the highest learning potential so far found. By training the student only on curated high-regret environments, PLR provably results in a minimax-regret student policy at Nash equilibrium. MAESTRO makes use of PLR-style curation of environment buffers to discover high-regret environments during training.

## 2.2 CO-PLAYER AUTOCURRICULA IN COMPETITIVE MULTI-AGENT RL

The choice of co-players plays a crucial role for training agents in multi-agent domains (Leibo et al., 2019). A simple but effective approach is *self-play* (SP, Silver et al., 2018), whereby the agent always plays with copies of itself. In competitive two-player games, SP produces a curriculum in which opponents match each other in skill level. However, SP may cause cycles in the strategy space when agents forget how to play against previous versions of their policy (Garnelo et al., 2021).

*Fictitious Self-Play* (FSP, Heinrich et al., 2015) overcomes the emergence of such cycles by training an agent against a uniform mixture of all previous policies. However, FSP can result in wasting a large number of interactions against significantly weaker opponents.

*Prioritized Fictitious Self-Play* (PFSP, Vinyals et al., 2019) mitigates this potential inefficiency by matching agent A with a frozen opponent B from the set of candidates $\mathcal{C}$ with probability $\frac{f(\mathbb{P}[A \text{ beats } B])}{\sum_{C \in \mathcal{C}} f(\mathbb{P}[A \text{ beats } C])}$, where $f$ defines the exact curriculum over opponents. For example, $f_{hard}(x) = (1 - x)^p$, where $p \in \mathbb{R}_+$, forces PFSP to focus on the hardest opponents.

## 3 METHOD

### 3.1 AN ILLUSTRATIVE EXAMPLE

To highlight the importance of curricula over the joint space of environments and co-players, we provide an illustrative example of a simple two-player game in Table 1. Here, the goal of the regret-maximising teacher is to select an environment/co-player pair for the student. Ignoring the co-player and selecting the highest regret environment leads to choosing $\theta_3$. Similarly, ignoring environments and selecting the highest regret co-player leads to $\pi_C$. This yields a suboptimal pair $(\theta_3, \pi_C)$ with REGRET$(\theta_3, \pi_C) = 0.4$, whereas a teacher over the joint space yields the optimal pair $(\theta_1, \pi_A)$ with

Table 1: **An illustrative two-player game.** Rows correspond to co-player policies and $\Pi = \{\pi_A, \pi_B, \pi_C\}$. Columns indicate different environments and $\Theta = \{\theta_1, \theta_2, \theta_3, \theta_4\}$. The payoff matrix represents the regret of the student on pair $(\theta, \pi)$.

| | $\theta_1$ | $\theta_2$ | $\theta_3$ | $\theta_4$ | $\mathbb{E}_{\theta \sim \Theta}$ |
|---|---|---|---|---|---|
| $\pi_A$ | 0.6 | 0.1 | 0.4 | 0.2 | 0.325 |
| $\pi_B$ | 0.1 | 0.5 | 0.4 | 0.3 | 0.325 |
| $\pi_C$ | 0.2 | 0.4 | 0.4 | 0.4 | 0.35 |
| $\mathbb{E}_{\pi \sim \Pi}$ | 0.3 | 0.33 | 0.4 | 0.3 | |

REGRET$(\theta_1, \pi_A) = 0.6$. Thus, naively treating the environment and co-player as independent can yield a sub-optimal curriculum. Such overspecialisation toward a subset of environmental challenges at the expense of overall robustness commonly emerges in multi-agent settings (Garnelo et al., 2021).

### 3.2 MULTI-AGENT ENVIRONMENT DESIGN STRATEGIST FOR OPEN-ENDED LEARNING

In this section, we describe a new multi-agent UED approach called ***Multi-Agent Environment Design Strategist for Open-Ended Learning*** (MAESTRO) which induces a regret-based autocurricula jointly over environments and co-players.

MAESTRO is a replay-guided approach (Jiang et al., 2021a) that relies on an environment generator to continuously create new environment instances. Rather than storing a single environment buffer, as done by PLR, MAESTRO maintains a population of policies $\mathfrak{B}$ with each policy assigned its own environment buffer. In each environment $\theta$ produced by the generator, MAESTRO evaluates the student's performance against a non-uniform mixture of co-player policies in $\mathfrak{B}$. High-regret environment/co-player pairs are

---

**Algorithm 1: MAESTRO**

1 **Input:** Environment generator $\Theta$
2 **Initialise:** Student policy $\pi$, co-player population $\mathfrak{B}$
3 **Initialise:** Environment buffers $\forall \pi' \in \mathfrak{B}, \Lambda(\pi') := \emptyset$.
4 **for** $i = \{1, 2, \dots\}$ **do**
5   **for** *many episodes* **do**
6     $\pi' \sim \mathfrak{B}$       ▷ Sample co-player via Eq 1
7     Sample replay decision     ▷ see Section 3.2.2
8     **if** *replaying* **then**
9       $\theta \sim \Lambda(\pi')$   ▷ Sample a replay environment
10       Collect trajectory $\tau$ of $\pi$ using $(\theta, \pi')$
11       Update $\pi$ with rewards $\boldsymbol{R}(\tau)$
12     **else**
13       $\theta \sim \Theta$   ▷ Sample a random environment
14       Collect trajectory $\tau$ of $\pi$ using $(\theta, \pi')$
15     Compute regret score $S = \widetilde{Regret}(\theta, \pi')$
16     Update $\Lambda(\pi')$ with $\theta$ using score $S$
17   $\mathfrak{B} \leftarrow \mathfrak{B} \cup \{\pi_i^\perp\}, \Lambda(\pi_i^\perp) := \emptyset$   ▷ frozen weights

---

then stored, along with the student agent's regret estimate for that pair, in the corresponding co-player's environment buffer.

MAESTRO maintains a dynamic population of co-players of various skill levels throughout training (Czarnecki et al., 2020). Figure 1 presents the overall training paradigm of MAESTRO and Algorithm 1 provides its pseudocode. We note that when applied to a fixed singleton environment, MAESTRO becomes a variation of PFSP (Vinyals et al., 2019), where the mixture of policies from the population is computed based on the agent's regret estimate, rather than the probability of winning.

### 3.2.1 MAINTAINING A POPULATION OF CO-PLAYERS

A key issue of using replay-guided autocurricula for multi-agent settings is nonstationarity. Specifically, using PLR with SP results in inaccurate regret estimates over environment/co-player pairs, as the co-player policies evolve over training. MAESTRO overcomes this issue by maintaining a population of past policies (Lanctot et al., 2017). This approach confers several key benefits. First, re-encountering past agents helps avoid cycles in strategy space (Balduzzi et al., 2019; Garnelo et al., 2021). Second, MAESTRO maintains accurate regret estimates in the face of nonstationarity by employing a separate environment buffer for each policy in the population. Third, MAESTRO always optimises a *single* policy throughout training, rather than a set of distinct policies which can be computationally expensive.

### 3.2.2 CURATING THE ENVIRONMENT/CO-PLAYER PAIRS

To optimise for the global regret over the joint environment/co-player space, a MAESTRO student is trained to best respond to a non-uniform mixture of policies from a population by prioritising training against co-players with high-regret environments in their buffers:

$$\text{CO-PLAYER}^{\text{HR}} \in \arg\max_{\pi' \in \mathfrak{B}} \{ \max_{\theta \in \Lambda(\pi')} \widetilde{Regret}(\theta, \pi') \}, \tag{1}$$

where $\mathfrak{B}$ is the co-player population, $\Lambda(\pi')$ is the environment buffer of agent $\pi'$, and $\widetilde{Regret}$ is the estimated regret of student for the pair $(\theta, \pi')$. To ensure that the student learns to best respond to high-regret co-players as well as to the entire population $\mathfrak{B}$, we enforce all members of $\mathfrak{B}$ to be assigned a minimum probability $\frac{\lambda}{N}$. For instance, if Equation (1) returns a single highest-regret co-player, then the resulting prioritised distribution assigns a weight of $\frac{N-\lambda(N-1)}{N}$ to the highest-regret co-player and weight of $\frac{\lambda}{N}$ to the remaining co-players. For each co-player $\pi' \in \mathfrak{B}$, MAESTRO maintains a PLR environment buffer $\Lambda(\pi')$ with the top-$K$ high-regret levels. Once the co-player is sampled, we make a replay decision: with probability $p$, we use a bandit to sample a training environment from $\Lambda(\pi')$,[2] and with probability $1 - p$, we sample a new environment for evaluation from the environment generator. Similar to Robust PLR (Jiang et al., 2021a), we only update the student policy on environments sampled from the environment buffer. This provides MAESTRO with strong robustness guarantees, which we discuss in Section 3.3.

### 3.3 ROBUSTNESS GUARANTEES OF MAESTRO

We analyse the expected behaviour of MAESTRO if the system reaches equilibrium: does MAESTRO produce a regret-maximising distribution over environments and co-players and is the policy optimal with respect to these distributions? We cast this problem in terms of Bayesian Nash equilibrium (BNE) behaviour in individual environments. BNE is an extension of Nash equilibrium (NE) where each co-player $j$ has an unknown *type* parameter $\theta^j$, which affects the dynamics of the game and is only known to that player. The distribution over these parameters $\tilde{\Theta}^N$ is assumed to be common knowledge. Equilibria are then defined as the set of policies, conditioned on their unknown type, each being a best response to the policies of the other players. That is, for policy $\pi_j$ of any player $j$,

$$\pi_j \in \arg\max_{\hat{\pi}_j \in \Pi_j} \{ \mathbb{E}_{\theta^N \in \tilde{\Theta}^N} [U(\hat{\pi}_j(\theta^j), \boldsymbol{\pi}_{-j}(\theta^{-j}))] \}. \tag{2}$$

In MAESTRO, we can assume each co-player is effectively omniscient, as each is co-evolved for maximal performance against the student in the environments it is paired with in its high-regret environment buffer. In contrast, the student has conferred no special advantages and has access to only the standard observations. We formalise this setting as a $-i$-knowing game. This game corresponds to the POSG with the same set of players, action space, rewards, and states as the original UPOSG, but with $\theta$ sampled at the first time step and provided to co-players $-i$ as part of their observation.

---

[2]Sampling is based on environment's regret score, staleness, and other features following (Jiang et al., 2021b).

**Definition 1.** *The -i-knowing-game of an UPOSG* $\mathcal{M} = \langle n, \mathcal{A}, \mathcal{O} = \times_{i \in N} \mathcal{O}_i, \Theta, S, \mathcal{T}, \mathcal{I} = \times_{i \in N} \mathcal{I}_i, \mathcal{R} = \times_{i \in N} \mathcal{R}_i, \gamma \rangle$ *with parameter distribution $\tilde{\theta}$ is defined to be the POSG $K = \langle n' = n, \mathcal{A}' = \mathcal{A}, \mathcal{O}'_i = \mathcal{O}_i + \{\Theta \text{ if } i \in -i\}, S' = S, \mathcal{T}' = \mathcal{T}(\theta), \mathcal{I}'_i = \mathcal{I}_i + \{\theta \text{ if } \in -i\}, \mathcal{R}'_i = \mathcal{R}_i, \gamma \rangle$ where $\theta$ is sampled from the distribution $\tilde{\theta}$ on the first time step.*

We thus arrive at our main theorem, followed by a convenient and natural corollary for fully observable settings. We include the full proofs in Appendix A.

**Theorem 1.** *In two-player zero-sum settings, the* MAESTRO *student at equilibrium implements a Bayesian Nash equilibrium of the $-i$-knowing game, over a regret-maximising distribution of levels.*

**Corollary 1.** *In fully-observable two-player zero-sum settings, the* MAESTRO *student at equilibrium implements a Nash equilibrium in each environment in the support of the environment distribution.*

Informally, the proof of the Corollary 1 follows from the observation that the $-i$-knowing game in a fully observable setting is equivalent to the original distribution of environments, as there is no longer an information asymmetry between the student and co-players. Moreover, the NE strategy on this distribution of environment instances would be a NE strategy on each instance individually, given that they are fully observable. This argument is formalised in Appendix A.

## 4 EXPERIMENTAL SETTING

Our experiments aim to understand (1) the interaction between autocurricula over environments and co-players in multi-agent UED, (2) its impact on zero-shot transfer performance of student policies to unseen environments and co-players, and (3) the emergent complexity of the environments provided to the student agent under autocurricula. To this end, we evaluate methods in two distinct domains: discrete control with sparse rewards, and continuous control with dense rewards. We assess student robustness in OOD human-designed environments against previously unseen opponents. Given its strong performance and usage in related works, PPO (Schulman et al., 2017) serves as the base RL algorithm in our experiments. We provide full environment descriptions in Appendix B and detail our model architecture and hyperparameter choices in Appendix C.

**Baselines and Ablations** We compare MAESTRO against two key baselines methods producing autocurricula over environments: domain randomization (DR; Jakobi, 1997), and (Robust) PLR (Jiang et al., 2021a), a state-of-the-art UED baseline. For co-player curricula, we consider SP, FSP, and PFSP, popular methods that underlie breakthroughs such as AlphaGo (Silver et al., 2016) and AlphaStar (Vinyals et al., 2019). Since these baselines independently produce curricula either over environments or over co-players, our choice of baselines results in a combination of 6 joint curriculum baselines over the environment/co-player space. We present further ablations investigating the importance of MAESTRO's co-player selection mechanism in Appendix D.

### 4.1 ENVIRONMENTS

**LaserTag** is a grid-based, two-player zero-sum game proposed by Lanctot et al. (2017) where agents aim to tag each other with laser beams. Success in LaserTag requires agents to master sophisticated behaviours, including chasing opponents, hiding behind walls, keeping clear of long corridors, and maze-solving. Each agent observes the $5 \times 5$ grid area in front of it and can turn right or left, move forward, and shoot. Upon tagging an opponent, the agent and the opponent receive a reward of $1$ and $-1$, respectively, and the episode terminates. LaserTag training environments are generated by randomly sampling grid size, wall locations, and the initial locations and directions of agents.

**MultiCarRacing** (MCR, Schwarting et al., 2021) is a high-dimensional, pixel-based continuous control environment with dense rewards. Two agents compete by driving a full lap around the track. Each track consists of $n$ tiles. Agents receive a reward of $1000/n$ or $500/n$ for reaching a tile first or second, respectively. Agents receive a top-down, egocentric $96 \times 96 \times 3$ partial observation and can collide, push, or block each other, allowing for complex emergent behaviour and non-transitive dynamics. The action space is 3-dimensional, controlling changes in acceleration, braking, and steering. All training tracks used for training agents are generated by sampling 12 random control points defining a Bézier curve forming the track.

Figure 2 shows example LaserTag and MCR environments, including human-designed OOD test environments. MCR test environments are the Formula 1 CarRacing tracks from (Jiang et al., 2021a).

# 5 RESULTS AND DISCUSSION

## 5.1 CROSS-PLAY RESULTS

To assess the robustness of the approaches, we evaluate all pairs of methods in cross-play on OOD human-designed environments (13 LaserTag levels and 21 F1 tracks for MCR). For each pair of methods, we perform cross-play between all pairs of random seeds ($10 \times 10$ for LaserTag and $5 \times 5$ for MCR) on all environments. Full results are included in Appendix E.2.

Figure 3 shows the cross-play results on LaserTag throughout and at the end of training. To estimate the robustness of each method on unseen environments, we evaluate round-robin (RR) tournament results where each baseline is matched against every other baseline. More robust agents should attain higher RR returns across all other agents. We can see that MAESTRO outperforms all baselines in round-robin returns. Although SP-based methods achieve an early performance lead due to more frequent policy updates, MAESTRO quickly outperforms them due to its curriculum that prioritises challenging environment/opponent pairs.

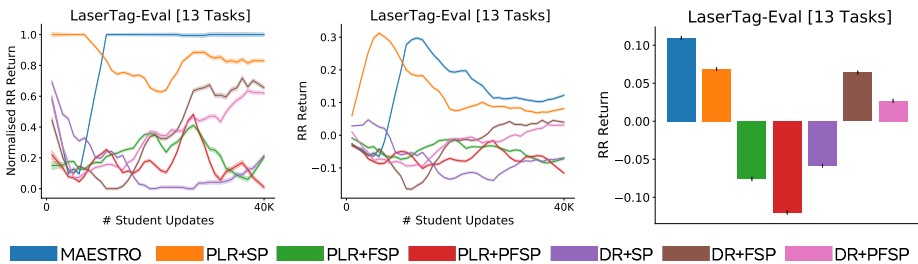

Figure 3: **LaserTag Cross-Play Results**. (Left) normalised and (Middle) unnormalised RR returns during training. (Right) RR returns at the end of training (mean and standard error over 10 seeds).

Figure 4 reports the MCR results on the challenging F1 tracks. Here, MAESTRO policies produce the most robust agents, outperforming all baselines individually across all tracks while also spending less time on the grass area outside of track boundaries. PLR-based baselines outperform DR, underscoring the benefits of a curriculum over environments on this task. Nonetheless, MAESTRO's superior performance highlights the importance of a joint curriculum over environments and co-players.

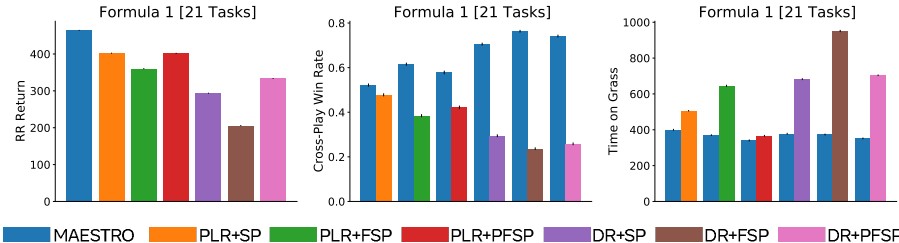

Figure 4: **MultiCarRacing Cross-Play Results**. (Left) RR returns (Middle) Cross-play win rate and (Right) grass time between MAESTRO and baselines (mean and standard error over 5 seeds).

## 5.2 IMPORTANCE OF THE CURRICULUM OVER THE ENVIRONMENT/CO-PLAYER SPACE

Figure 5 shows the co-player×environment regret landscape of a MAESTRO student agent on randomly-sampled environments against each co-player from MAESTRO's population at the end of training (as done in the motivating example in Section 3.1). We observe that high regret estimates depend on both the choice of the environment and the co-player. Most importantly, maximising mean regrets over environments and co-players independently does not lead to maximum regret over the joint space, highlighting the importance of the curricula over the joint environment/co-player space.

## 5.3 EVALUATION AGAINST SPECIALIST AGENTS

While we show that MAESTRO outperforms baselines trained on a large number of environments, we are also interested in evaluating MAESTRO against *specialist* agents trained only on single, *fixed* environments. Figure 6 shows that, despite having never seen the target environments, the MAESTRO agent beats specialist agents trained exclusively on them for the same number of updates. This is

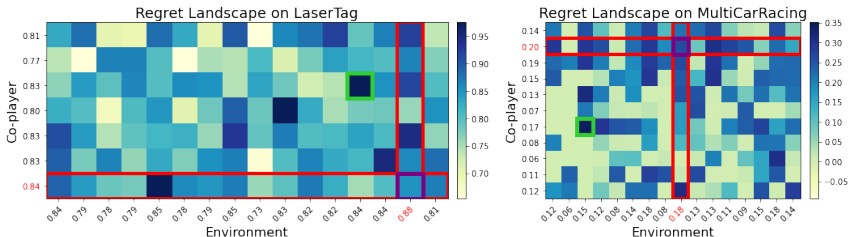

Figure 5: **Regret landscapes on LaserTag and MultiCarRacing.** Shown are regret estimates of the student on 16 random environment (columns) against each policy from MAESTRO's co-player population (rows). Highlighted in red are the regret estimates of the co-player and environment with the highest mean values when considered in isolation, whereas in green we highlight the overall highest regret environment/co-player pair.

possible because the MAESTRO agent was trained with an autocurriculum that promotes the robustness of the trained policy, allowing it to transfer better to unseen opponents on OOD environments. These results demonstrate the potential of multi-agent UED in addressing key problems in multi-agent RL, such as exploration (Leonardos et al., 2021) and co-player overfitting (Lanctot et al., 2017).

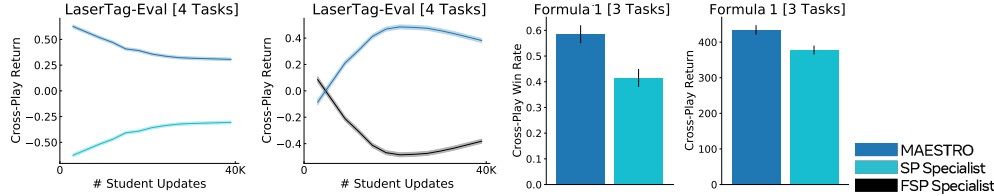

Figure 6: **Cross-Play vs Specialists**. Cross-play evaluation against specialist agents trained directly on the target task. We use different specialist agents for each fixed environment. Shown are averaged metrics across 4 target LaserTag levels and 3 Formula 1 tracks. Full results on individual environments are included in Appendix E.1.

## 5.4 EMERGENT CURRICULUM ANALYSIS

We analyse the curriculum over environments induced by MAESTRO and compare it with the curricula generated by the baselines. Figure 7 demonstrates that, as training progresses, MAESTRO provides agents with LaserTag environments with an increasing density of walls, which contributes to a gradual increase in environment complexity. Initially, the size of the grid also increases as the teacher challenges the student to navigate the mazes. However, after more adept opponent agents enter the population, larger mazes yield lower regret. Instead, MAESTRO starts prioritising smaller grid sizes compared to other baselines. Smaller environments with high wall density challenge the student to navigate maze structures while succeeding in more frequent multi-agent interactions against increasingly more capable opponents throughout an episode. Figure 2c illustrates such challenging environments, which, due to the smaller grid, require the student to engage with the opponent earlier in the episode. PLR-based methods also prioritise high wall density but do not shrink the grid size to prioritise competitive interactions to the same extent as MAESTRO.

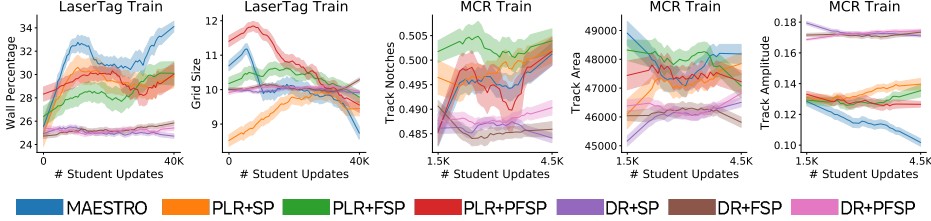

Figure 7: **Characteristics of emergent autocurricula in LaserTag and MultiCarRacing.**

We observe similar emergent complexity in the MCR domain. Here the PLR-based baselines and MAESTRO all prioritise tracks that have a high number of notches (i.e. the non-convex parts of the track) and enclosed areas. Here, as in the LaserTag domain, MAESTRO gradually shrinks the track amplitude, which corresponds to a lower intensity of the winding portions of the track. Such tracks may contain more segments where competitive interactions can play out without sacrificing overall episodic return, thus inducing policies more adept at the competitive aspects of this domain. Figure 2 illustrates randomly sampled environments curated by MAESTRO at different stages of training.

## 6 Related Work

*Unsupervised Environment Design* (UED, Dennis et al., 2020) is a family of methods that provide an agent with a sequence of environments for training robust policies. The simplest UED approach is *Domain Randomisation* (DR, Jakobi, 1997; Sadeghi & Levine, 2017) which has demonstrated strong empirical performances in domains such as robotics (Tobin et al., 2017; James et al., 2017) and magnetic control of tokamak plasmas (Degrave et al., 2022). PAIRED (Dennis et al., 2020; Gur et al., 2021) trains an environment generator that maximises the student's regret, approximated as the difference in return between the student and an antagonist agent. *Prioritized Level Replay* (PLR, Jiang et al., 2021a;b) curates environment instances (i.e., levels) for training, by performing a random search of domain randomised levels for those with high learning potential, e.g., as measured by estimated regret. *ACCEL* (Parker-Holder et al., 2022) is a replay-guided UED approach that extends PLR by making edits to high-regret environments. Several methods generate curricula by adapting the environment parameters in response to the agent's performance (Portelas et al., 2019; Matiisen et al., 2020; Klink et al., 2019; Eimer et al., 2021). This adaptation is largely heuristic-driven, without the robustness guarantees shared by minimax-regret UED methods. Notably, all these methods focus on single-agent RL, while MAESTRO is designed for the two-player multi-agent setting.

Many prior works study curricula over opponents in two-player zero-sum settings. The most naive approach, self-play (SP), consists of pitting the agent against a copy of itself. Combined with search, SP has led to superhuman performances in board games such as Backgammon (Tesauro, 1995), Chess and Go (Silver et al., 2016). Zinkevich et al. (2007) use self-play with regret minimisation for achieving Nash equilibrium, an approach that led to superhuman performance in Poker (Brown & Sandholm, 2018; 2019). *Fictitious self-play* (FSP) learns a best-response to the uniform mixture of all previous versions of the agent (Brown, 1951; Leslie & Collins, 2006; Heinrich et al., 2015). *Prioritised fictitious self-play* (PFSP, Vinyals et al., 2019) trains agents against a non-uniform mixture of policies based on the probability of winning against each policy. PFSP is a practical variant of *Policy-Space Response Oracles* (PSRO, Lanctot et al., 2017), a general population learning framework, whereby new policies are trained as best responses to a mixture of previous policies. MAESTRO is related to PSRO but adapted for UPOSGs. In MAESTRO, the population meta-strategy is based on the student's regret when playing against policies on environments observed during training. Unlike our work, these prior autocurricula methods for competitive multi-agent environments do not directly consider variations of the environment itself.

Several prior works have applied DR in multi-agent domains. Randomly modifying the environment has proven critical for the emergence of complex behaviours in Hide-and-Seek (Baker et al., 2019), Capture the Flag (Jaderberg et al., 2019), and StarCraft II Unit Micromanagement (Ellis et al., 2022). In XLand (Open Ended Learning Team et al., 2021), a curriculum is provided over both environments and tasks to create general learners. This work differs from ours in multiple aspects. Open Ended Learning Team et al. (2021) uses handcrafted heuristics and rejection sampling for selecting environments for training and evaluating agents, while MAESTRO automatically selects environments based on regret rather than hand-coded heuristics. Furthermore, unlike the autocurricula used in XLand, MAESTRO does not rely on population-based training, a computationally expensive algorithm for tuning the autocurriculum hyperparameters.

## 7 Conclusion and Future Work

In this paper, we provided the first formalism for multi-agent learning in underspecified environments. We introduced MAESTRO, an approach for producing an autocurriculum over the joint space of environments and co-players. Moreover, we proved that MAESTRO attains minimax-regret robustness guarantees at Nash equilibrium. Empirically MAESTRO produces agents that are more robust to the environment and co-player variations than a number of strong baselines in two challenging domains. MAESTRO agents even outperform specialist agents in these domains. Our work opens up many interesting directions for future work. MAESTRO could be extended to $n$-player games, as well as co-operative and mixed settings. Furthermore, MAESTRO could be combined with search-based methods in order to further improve sample efficiency and generalisation. Another interesting open question is identifying conditions whereby such an algorithm can provably converge to Nash equilibrium in two-player zero-sum settings. Finally, this work is limited to training only a single policy for each of the multi-agent UED approaches. Concurrent and continued training of several unique policies in underspecified multi-agent problems could be a generally fruitful research direction.

## ACKNOWLEDGEMENTS

We extend our gratitude to Christopher Bamford for his assistance with the Griddly sandbox framework (Bamford et al., 2020; 2022) used for our LaserTag experiments. We thank Max Jaderberg, Marta Garnelo, Edward Grefenstette, Yingchen Xu, and Robert Kirk for insightful discussions and valuable feedback on this work. We also thank the anonymous reviewers for their recommendations on improving the paper. This work was funded by Meta AI.

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

# A  THEORETICAL RESULTS

It is useful to understand the long-term behaviour of MAESTRO the student and teacher agents reach optimality. In this section, we will formally characterise this equilibrium behaviour, showing MAESTRO achieves a Bayesian Nash equilibrium in a modified game, which we call the $-i$-knowing game, in which the other player has prior knowledge of the environment and can design their policy accordingly. We will do this by first defining the $-i$-knowing game, showing that the policies the MAESTRO student learns in equilibria represent Bayesian Nash equilibrium strategies in this game, and then specialising this result to a corollary focused on fully observable games where MAESTRO finds a Nash equilibrium for all environments in support of the teacher distribution.

We will define the $-i$-knowing-game, given a UPOSG $\mathcal{M}$, with a parameter distribution $\tilde{\theta}$ as POSG constructed as a modification of the original game where the distribution over parameters is determined by $\tilde{\theta}$, and the agents other than $i$ know the true parameters of the world. This simulates the setting where the co-player is a specialist for the particular environment, who has played the game many times. More formally:

**Definition 1.** *The -i-knowing-game of an UPOSG* $\mathcal{M} = \langle n, \mathcal{A}, \mathcal{O} = \times_{i \in N} \mathcal{O}_i, \Theta, S, \mathcal{T}, \mathcal{I} = \times_{i \in N} \mathcal{I}_i, \mathcal{R} = \times_{i \in N} \mathcal{R}_i, \gamma \rangle$ *with parameter distribution* $\tilde{\theta}$ *is defined to be the POSG* $K = \langle n' = n, \mathcal{A}' = \mathcal{A}, \mathcal{O}_i' = \mathcal{O}_i + \{\Theta \text{ if } i \in -i\}, S' = S, \mathcal{T}' = \mathcal{T}(\theta), \mathcal{I}_i' = \mathcal{I}_i + \{\theta \text{ if } \in -i\}, \mathcal{R}_i' = \mathcal{R}_i, \gamma \rangle$ *where* $\theta$ *is sampled from the distribution* $\tilde{\theta}$ *on the first time step. That is, a POSG with the same set of players, action space, rewards, and states as the original UPOSG, but with* $\theta$ *sampled once at the beginning of time, fixed into the transition function and given to the agents* $-i$ *as part of their observation.*

We use $U^K(\pi_i; \pi_{-i}^K; \tilde{\theta})$ to refer to the utility function in the $-i$-knowing game, which can be written in terms of the utility function of the original UPOSG as

$$U^K(\pi_i; \pi_{-i}^K; \tilde{\theta}) = \mathbb{E}_{\theta \sim \tilde{\theta}}[U(\pi_i; \pi_{-i}^K(\theta), \theta)],$$

where $\pi_{-i}^K(\theta)$ is the policy for players $-i$ in the $-i$-knowing game conditioned on $\theta$. Given this definition, we can prove the main theorem, that the equilibrium behaviour of MAESTRO represents a Bayesian Nash equilibrium of this game.

**Theorem 1.** *In two-player zero-sum settings, the* MAESTRO *student at equilibrium implements a Bayesian Nash equilibrium of the* $-i$-*knowing game, over a regret-maximising distribution of levels.*

*Proof.* Let $\pi_i, \tilde{\theta}^M$ be a pair which is in equilibrium in the MAESTRO game. That is:

$$\pi_i \in \arg\max_{\pi_i \in \Pi_i} \{ \mathbb{E}_{\theta, \pi_{-i} \sim \tilde{\theta}^M}[U(\pi_i; \pi_{-i}; \theta)] \} \tag{3}$$

$$\tilde{\theta}^M \in \arg\max_{\tilde{\theta}^M \in \Delta(\Theta \times \Pi_{-i})} \{ \mathbb{E}_{\theta, \pi_{-i} \sim \tilde{\theta}^M}[U(\pi_i^*; \pi_{-i}; \theta) - U(\pi_i; \pi_{-i}; \theta)] \} \tag{4}$$

where $\pi_i^*$ is an optimal policy for player $i$ given $\pi_{-i}$ and $\theta$, while $\Delta(S)$ denotes the set of distributions over S. Then we can define $D^{Regret}$ to be the marginal distribution over $\theta$ from samples $\theta, \pi_{-i} \sim \tilde{\theta}^M$. Define $\pi_{-i}^K(\theta)$ as the marginal distribution over $\pi_{-i}$ sampled from $\tilde{\theta}^M$ conditioned on $\theta$ for $\theta$ in the support of $\tilde{\theta}^M$ and $\pi_{-i}$ a best response to $\theta$ and $\pi_i$ otherwise.

We will show that $(\pi_i; \pi_{-i}^K)$ is a Bayesian Nash equilibrium on $-i$-knowing game of $\mathcal{M}$ with a regret-maximizing distribution over parameters $D^{Regret}$. We can show both of these by unwrapping and re-wrapping our definitions.

First to show $\pi_i \in \arg\max_{\pi_i \in \Pi_i} \{ \mathbb{E}_{\tilde{\theta} \sim D^{Regret}}[U^K(\pi_i; \pi_{-i}^K; \tilde{\theta})] \}$:

$$\pi_i \in \arg\max_{\pi_i \in \Pi_i} \{ \mathbb{E}_{\tilde{\theta} \sim D^{Regret}}[U^K(\pi_i; \pi_{-i}^K; \tilde{\theta})] \} \tag{5}$$

$$\iff \pi_i \in \arg\max_{\pi_i \in \Pi_i} \{ \mathbb{E}_{\theta, \pi_{-i} \sim \tilde{\theta}^M}[U(\pi_i; \pi_{-i}^K(\theta); \theta)] \} \tag{6}$$

$$\iff \pi_i \in \arg\max_{\pi_i \in \Pi_i} \{ \mathbb{E}_{\theta, \pi_{-i} \sim \tilde{\theta}^M}[U(\pi_i; \pi_{-i}; \theta)] \} \tag{7}$$

Which is known by the definition of Nash equilibrium in the MAESTRO game, in Equation 3.

Similarly, we show that we have $\pi_{-i}^K \in \arg\max\{\underset{\tilde{\theta} \sim D^{Regret}}{\mathbb{E}}[U^K(\pi_i; \pi_{-i}^K; \tilde{\theta})]\}$ by:

$$\pi_{-i}^K \in \underset{\pi_{-i}^K \in \Pi_{-i}^K}{\arg\max}\{\underset{\tilde{\theta} \sim D^{Regret}}{\mathbb{E}}[U^K(\pi_i; \pi_{-i}^K; \tilde{\theta})]\} \tag{8}$$

$$\iff \pi_{-i}^K \in \underset{\pi_{-i}^K \in \Pi_{-i}^K}{\arg\max}\{\underset{\theta, \pi_{-i} \sim \tilde{\theta}^M}{\mathbb{E}}[U(\pi_i; \pi_{-i}^K(\theta); \theta)]\} \tag{9}$$

$$\iff \pi_{-i}^K(\theta) \in \underset{\pi_{-i}^K \in \Pi_{-i}^K}{\arg\max}\{\underset{\theta, \pi_{-i} \sim \tilde{\theta}^M}{\mathbb{E}}[U(\pi_i; \pi_{-i}; \theta)]\}. \tag{10}$$

The final line of which follows from the fact that $\pi_{-i}$ is a best-response $\pi_i$ for each $\theta$. More concretely, this can be seen in Equation 4 by noting that $\theta$ and $\pi_{-i}$ conditioned on a specific $\theta$ can be independently optimised and holding $\theta$ fixed. □

Using this theorem, we can also prove a natural and intuitive corollary for the case where the environment is fully observable:

**Corollary 1.** *In fully-observable two-player zero-sum settings, the* MAESTRO *student at equilibrium implements a Nash equilibrium in each environment in the support of the environment distribution.*

*Proof.* From Theorem 1 we have:

$$\pi_i \in \underset{\pi_i \in \Pi_i}{\arg\max}\{\underset{\tilde{\theta} \sim D^{Regret}}{\mathbb{E}}[U^K(\pi_i; \pi_{-i}^K; \tilde{\theta})]\}$$

However, since the world is fully observable, the policy $\pi_{-i}^K$ can be made independent of the additional observation $\theta$ in the $-i$-knowing game since that can be inferred from the information already in the agent's observations. As such, $\pi_{-i}^K$ can be interpreted as a policy in the original game, giving:

$$\pi_i \in \underset{\pi_i \in \Pi_i}{\arg\max}\{\underset{\theta \sim D^{Regret}}{\mathbb{E}}U(\pi_i; \pi_{-i}^K(\theta); \theta)\}$$

Moreover, since the environment is fully observable, $\pi_i$ can condition on $\theta$, so for it to be optimal for the distribution, it must be optimal for each level in the support of the distribution. Giving, for each $\theta$ be in the support of $\tilde{\theta}^M$:

$$\pi_i \in \underset{\pi_i \in \Pi_i}{\arg\max}\{U(\pi_i; \pi_{-i}^K(\theta); \theta)\},$$

showing that $\pi$ is a best-response to $\pi_{-i}^K(\theta)$ on $\theta$. The same arguments can be followed to show that $\pi_{-i}^K(\theta)$ is a best response to $\pi$ on $\theta$. Since each policy is a best response to the other, they are in a Nash equilibrium as desired. □

Thus, if MAESTRO reaches an equilibrium in a fully observable two-player zero-sum setting, it behaves as expected by achieving a Nash equilibrium in every environment in support of the curriculum distribution $\tilde{\theta}^M$.

# B  ENVIRONMENT DETAILS

This section describes the environment-specific details used in our experiments. For both LaserTag and MultiCarRacing, we outline the process of environment generation, present held-out evaluation environments, as well as other relevant information.

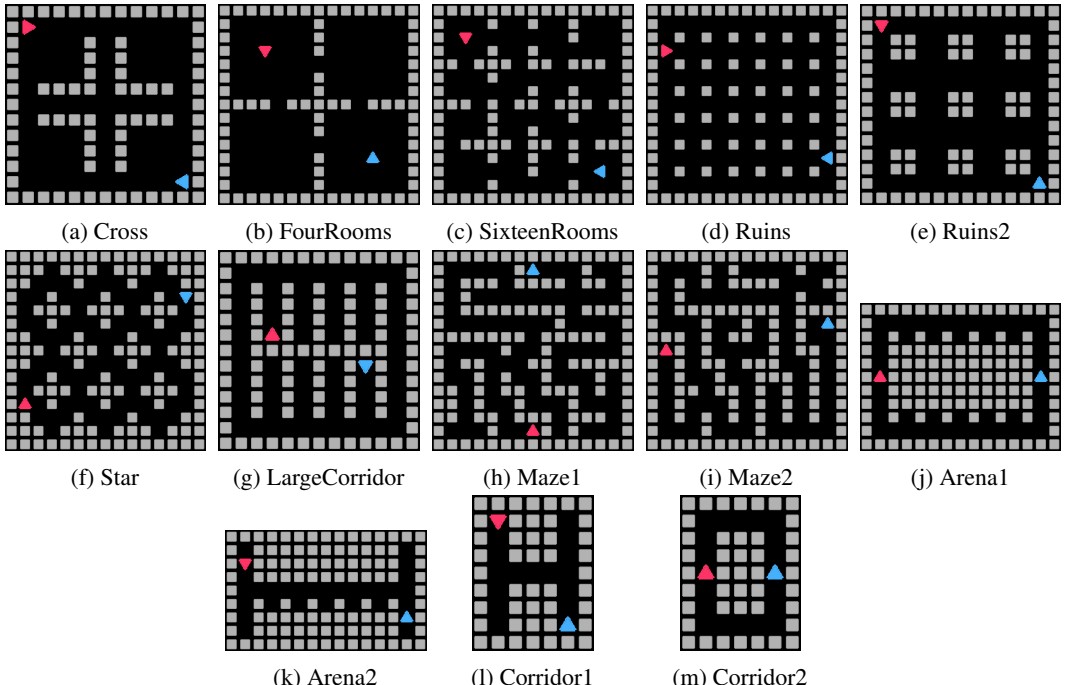

Figure 8: Evaluation environments for LaserTag.

### B.1 LASERTAG

LaserTag is a two-player zero-sum grid-based game, where two agents aim to tag each other with a light beam under partial observability. It is inspired by prior singleton variation used in (Lanctot et al., 2017; Leibo et al., 2017) and developed using the Griddly sandbox framework (Bamford et al., 2020; 2022). LaserTag challenges the agent to master various sophisticated behaviour, such as chasing opponents, hiding behind walls, keeping clear of long corridors, maze solving, etc. Each agent can only observe a $5 \times 5$ area of the grid in front of it. The action space includes the following 5 actions: turn right, turn left, move forward, shoot, and no-op. Upon tagging an opponent, an agent receives a reward of 1, while the opponent receives a $-1$ penalty, after which the episode is restarted. If the episode terminates while the two agents are alive, neither of the agents receives any reward.

All environment variations (or levels) that are used to train agents are procedurally generated by an environment generator. Firstly, the generator samples the size of the square grid (from $5 \times 5$ to $15 \times 15$) and the percentage of the walls in it (from $0\%$ to $50\%$) uniformly at random. Then the generator samples random locations for the walls, followed by the locations and directions of the two agents. Figure 2 illustrates some levels sampled from the level generator. Note that the generator can generate levels where agents are unreachable.

Upon training the agents on randomly generated levels, we assess their robustness on previously unseen human-designed levels shown in Figure 8 against previously unseen agents.

### B.2 MULTICARRACING

MultiCarRacing is a continuous control problem with dense rewards and pixel-based observations (Schwarting et al., 2021). Each track consists of $n$ tiles, with cars receiving a reward of $1000/n$ or $500/n$, depending on if they reach the tile first or second respectively. An additional penalty of $-0.1$ is applied at every timestep. Episodes finish when all tiles have been driven over by at least one car. If a car drives out of bounds of the map (rectangle area encompassing the track), the car "dies" and the episode is terminated. Each agent receives a $96 \times 96 \times 3$ image as observation at each timestep. The action space consists of 3 simultaneous moves that change the gas, brakes, and steering direction

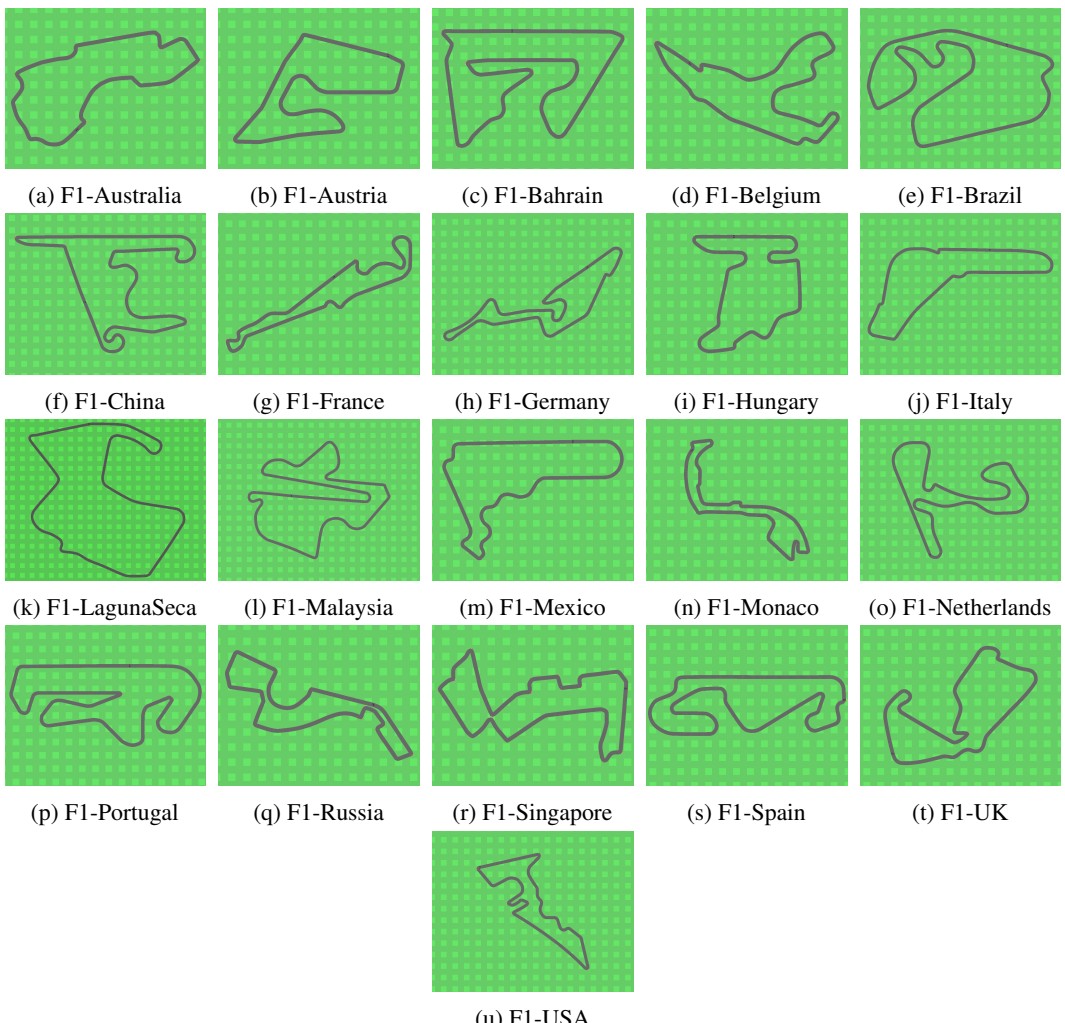

Figure 9: Evaluation Formula 1 tracks for MultiCarRacing originally from Jiang et al. (2021a).

of the car. For this environment, we recognise the agent with a higher episodic return as the winner of that episode.

All tracks used to train student agents are procedurally generated by an environment generator, which was built on top of the original MultiCarRacing environment (Schwarting et al., 2021). Each track consists of a closed loop around which the agents must drive a full lap. In order to increase the expressiveness of the original MultiCarRacing, we reparameterized the tracks using Bézier curves. In our experiments, each track consists of a Bézier curve based on 12 randomly sampled control points within a fixed radius of $B/2$ of the centre $O$ of the playfield with $B \times B$ size.

For training, additional reward shaping was introduced similar to (Ma, 2019): an additional reward penalty of $-0.1$ for driving on the grass, a penalty of $-0.5$ for driving backwards, as well as an early termination if cars spent too much time on grass. These are all used to help terminate less informative episodes. We utilise a memory-less agent with a frame stacking $= 4$ and with sticky actions $= 8$. After training the agents on randomly generated tracks, we assess their robustness on previously unseen 20 real-world Formula 1 (F1) tracks designed to challenge professional racecar drivers proposed by (Jiang et al., 2021a) and shown in Figure 9.

# C  IMPLEMENTATION DETAILS

In this section, we detail the agent architectures, hyperparameter choices, and evaluation procedures used in our experiments discussed in Section 4. We use PPO to train the student agent in all experiments. Table 2 summarises our final hyperparameter choices for all methods.

All experiments are performed on an internal cluster. Each job (representing a seed) is performed with a single Tesla V100 GPU and 10 CPUs. For each method, we train 10 LaserTag agents for approximately 7 days and 5 MultiCarRacing agents for approximately 15 days.

## C.1  LASERTAG

**Agent Architecture:** The student policy architecture is adapted from (Dennis et al., 2020; Jiang et al., 2021a). Our model encodes the partial grid observation using a convolution layer ($3 \times 3$ kernel, stride length 1, 16 filters) followed by a ReLU activation layer over the flattened convolution outputs. This is then passed through an LSTM with hidden dimension 256, followed by two fully-connected layers, each with a hidden dimension of size 32 with ReLU activations, to produce the action logits over the 5 possible actions. The model does not receive the agent's direction as input.

**Evaluation Procedure:** For each pair of baselines, we evaluate cross-play performance between all pairs of random seeds ($10 \times 10$ combinations) over 5 episodes on 13 human-designed LaserTag levels, resulting in a total of 6500 evaluation episodes for a given checkpoint.

**Choice of Hyperparameters:** Many of our hyperparameters are inherited from previous works such as (Dennis et al., 2020; Jiang et al., 2021b;a; Parker-Holder et al., 2022) with some small changes. We selected the best performing settings based on the average return on the unseen validation levels against previously unseen opponents on at least 5 seeds.

We conducted a coarse grid search over student learning rate in $\{5 * 10^{-4}, 10^{-4}, 5 * 10^{-5}, 10^{-5}\}$, number of minibatches per epoch in $\{1, 2, 4\}$, entropy coefficients in $\{0, 10^{-3}, 5*10^{-3}\}$, and number of epochs in $\{2, 5, 10\}$. For agent storage, we tested adding a copy of the student agent in the storage after every $\{2000, 4000, 6000, 8000\}$ student update. For PFSP, we compute the win rate between agents in the last 128 episodes. We further conducted a grid search over the entropy parameter of $f_{hard}$ in $\{1.5, 2, 3\}$ and a smoothing constant which adds a small value to each probability in PFSP with $\{0.1, 0.2\}$ values so that all previous checkpointed agents have a nonzero probability to be replayed again.[3] For the parameters of PLR, we conducted a grid search over level replay rate $p$ in $\{0.5, 0.9\}$, buffer size in $\{4000, 8000, 12000\}$, staleness coefficient $\rho$ in $\{0.3, 0.7\}$, as well as the level replay score functions in {MaxMC, PVL} (see Appendix C.3 for information on score functions in PLR). For MAESTRO, we evaluated the co-player exploration coefficients in $\{0.05, 0.1\}$, and per-agent environment buffer sizes in $\{500, 750, 1000, 1200\}$.

## C.2  MULTICARRACING

**Agent Architecture:** The student policy architecture is based on the PPO implementation in (Ma, 2019). The model utilises an image embedding module consisting of a stack of 2D convolutions with square kernels of sizes 2, 2, 2, 2, 3, 3, channel outputs of 8, 16, 32, 64, 128, 256, and stride lengths of 2, 2, 2, 2, 1, 1 respectively, resulting in an embedding of size 256. This is then passed through a fully-connected layer with a hidden size of 100, followed by a ReLU nonlinearity. Then, the output is fed through two separate fully-connected layers, each with a hidden size of 100 and an output dimension equal to the action dimension, followed by softplus activations. We then add 1 to each component of these two output vectors, which serve as the $\alpha$ and $\beta$ parameters respectively for the Beta distributions used to sample each action dimension. We normalize rewards by dividing rewards by the running standard deviation of returns so far encountered during the training.

**Evaluation Procedure:** For each pair of baselines, we evaluate cross-play performance between all pairs of random seeds ($5 \times 5$ combinations) over 5 episodes on 21 OOD Formula 1 tracks Jiang et al. (2021a), resulting in a total of 2625 evaluation episodes for a given checkpoint.

---

[3]Otherwise, if the student agent wins all the episodes in their first encounter against opponent B, B will have 0 probability of being selected again.

Table 2: Hyperparameters used for training each method in the LaserTag and MultiCarRacing environments.

| Parameter | LaserTag | MultiCarRacing |
|---|---|---|
| *PPO* | | |
| $\gamma$ | 0.995 | 0.99 |
| $\lambda_{\text{GAE}}$ | 0.95 | 0.9 |
| PPO rollout length | 256 | 125 |
| PPO epochs | 5 | 8 |
| PPO mini-batches per epoch | 4 | 4 |
| PPO clip range | 0.2 | 0.2 |
| PPO number of workers | 32 | 32 |
| Adam learning rate | 1e-4 | 1e-4 |
| Adam $\epsilon$ | 1e-5 | 1e-5 |
| PPO max gradient norm | 0.5 | 0.5 |
| PPO value clipping | yes | no |
| Return normalization | no | yes |
| Value loss coefficient | 0.5 | 0.5 |
| Student entropy coefficient | 0.0 | 0.0 |
| | | |
| *PLR* | | |
| Replay rate, $p$ | 0.5 | 0.5 |
| Buffer size, $K$ | 4000 | 8000 |
| Scoring function | MaxMC | PVL |
| Prioritization | rank | rank |
| Temperature, $\beta$ | 0.3 | 1.0 |
| Staleness coefficient, $\rho$ | 0.3 | 0.7 |
| | | |
| *FSP* | | |
| Agent checkpoint interval | 8000 | 400 |
| | | |
| *PFSP* | | |
| $f_{hard}$ entropy coef | 2 | 2 |
| Win rate episodic memory | 128 | 128 |
| | | |
| MAESTRO | | |
| $\lambda$ coef | 0.1 | 0.1 |
| Buffer size for $\mathfrak{B}$ members | 1000 | 1000 |

**Choice of Hyperparameters:** Many of our hyperparameters are inherited from (Jiang et al., 2021a) with some small changes. We conducted a limited grid search over student learning rate in $\{10^{-4}, 3 * 10^{-4}\}$, number of actors in $\{16, 32\}$, PPO rollout length in $\{125, 256\}$. For agent storage, we tested adding a copy of the student agent in the storage after every $\{200, 400\}$ student update. For PFSP, we compute the win rate between agents in the last 128 episodes, while recognising the agent with a higher episodic return as the winner. For the parameters of PLR, we conducted a grid search over level buffer size in $\{4000, 6000, 8000\}$, staleness coefficient $\rho$ in $\{0.3, 0.7\}$, as well as the level replay prioritisation in {rank, proportional} (Jiang et al., 2021a). For MAESTRO, we evaluated the co-player exploration coefficients in $\{0.05, 0.1\}$, and per-agent environment buffer sizes in $\{500, 1000\}$.

Given the poor performance of a random co-player on the MultiCarRacing domain, agents are added to co-player populations in MAESTRO as well as FSP- and PFSP-based baselines only after 400 PPO updates. All baselines are trained using SP until that point.

## C.3 REGRET APPROXIMATIONS

Jiang et al. (2021a) proposes the following two score functions to approximate regret in PLR.

*Maximum Monte Carlo (MaxMC)* mitigates some of the bias of the PVL by replacing the value target with the highest empirical return observed on the given environment variation throughout training. MaxMC ensures that the regret estimate does not depend on the agent's current policy. It takes the form of $(1/T) \sum_{t=0}^{T} R_{\max} - V(s_t)$.

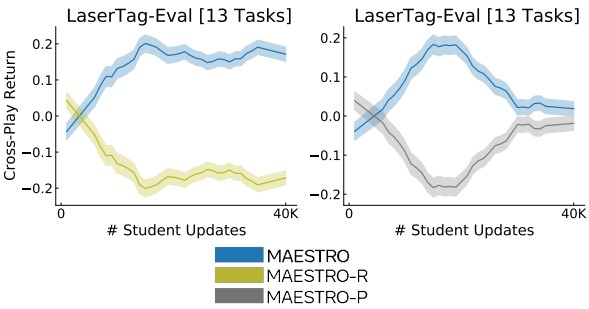

Figure 10: **Ablation Study Results**. Comparing MAESTRO against two variants: MAESTRO-R and MAESTRO-P. Plots show the mean and standard error across 10 training seeds.

*Positive Value Loss (PVL)* estimates the regret by computing the difference between the maximum achieved return and predicted return on an episodic basis. When GAE (Schulman et al., 2016) is used to estimate bootstrapped value targets, this loss takes the form of $\frac{1}{T} \sum_{t=0}^{T} \max \left( \sum_{k=t}^{T} (\gamma\lambda)^{k-t} \delta_k, 0 \right)$, where $\lambda$ and $\gamma$ are the GAE and MDP discount factors respectively, and $\delta_t$, the TD-error at timestep $t$.

In this work, we use MaxMC regret approximation in the LaserTag domain and PVL in MultiCarRacing.

## D    ABLATION STUDY

We perform an ablation study to evaluate the effectiveness of co-player selection in MAESTRO from a population based on their per-environment regret scores, as described in Equation (1). We consider different methods for selecting a co-player in MAESTRO (line 6 in Algorithm 1). MAESTRO-R samples the co-player uniformly at random, whilst MAESTRO-P uses PFSP's win rate heuristic for opponent prioritisation. Figure 10 illustrates that MAESTRO outperforms both variants in terms of sample-efficiency and robustness.

## E    FULL RESULTS

Agents are trained for $40000$ PPO updates on LaserTag and $4500$ PPO updates on MCR.

### E.1    MAESTRO VERSUS SPECIALISTS

Figures 11 and 12 show the cross-play performances between MAESTRO and specialist agents trained directly on the target environments for LaserTag and MultiCarRacing, respectively.

### E.2    CROSS-PLAY RESULTS

#### E.2.1    LASERTAG CROSS-PLAY

Figures 13 and 14 illustrate the round-robins returns with and without normalization between MAESTRO and other baselines throughout training on each held-out evaluation environment in LaserTag. Figure 15 shows the round-robin returns between MAESTRO and other baselines after the training.

#### E.2.2    MULTICARRACING CROSS-PLAY

Figure 16 illustrate the round-robin returns between MAESTRO and other baselines on each track of the Formula 1 benchmark (Jiang et al., 2021a). Figures 17, 18, and Figure 19 show the win rates, returns, and average time on grass during cross-play between MAESTRO and each baseline on Formula 1 benchmark.

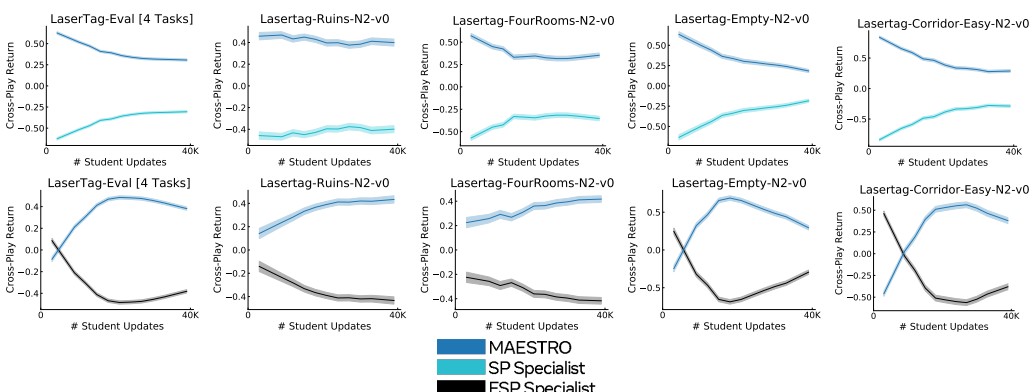

Figure 11: Cross-play between MAESTRO and specialist agents trained directly on the target environment in LaserTag.

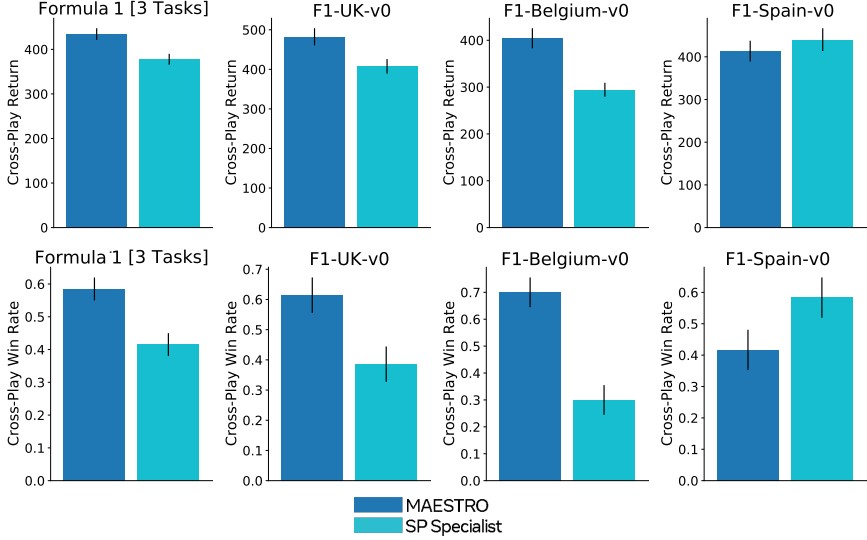

Figure 12: Cross-play between MAESTRO and specialist agents trained directly on the target environment in MultiCarRacing.

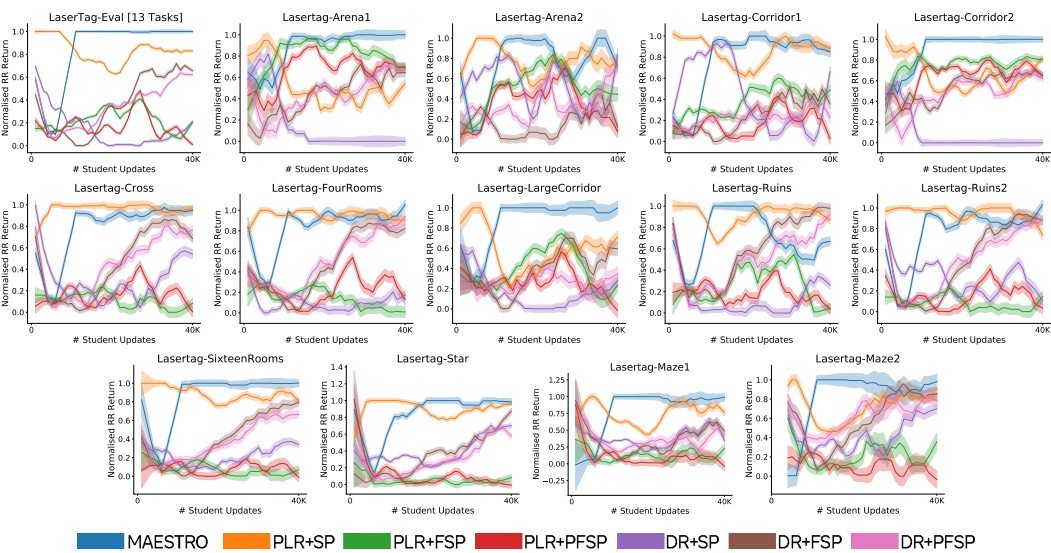

Figure 13: Normalised round-robin return in cross-play between MAESTRO and 6 baselines on all LaserTag evaluation environments throughout training (combined and individual). Plots show the mean and standard error across 10 training seeds.

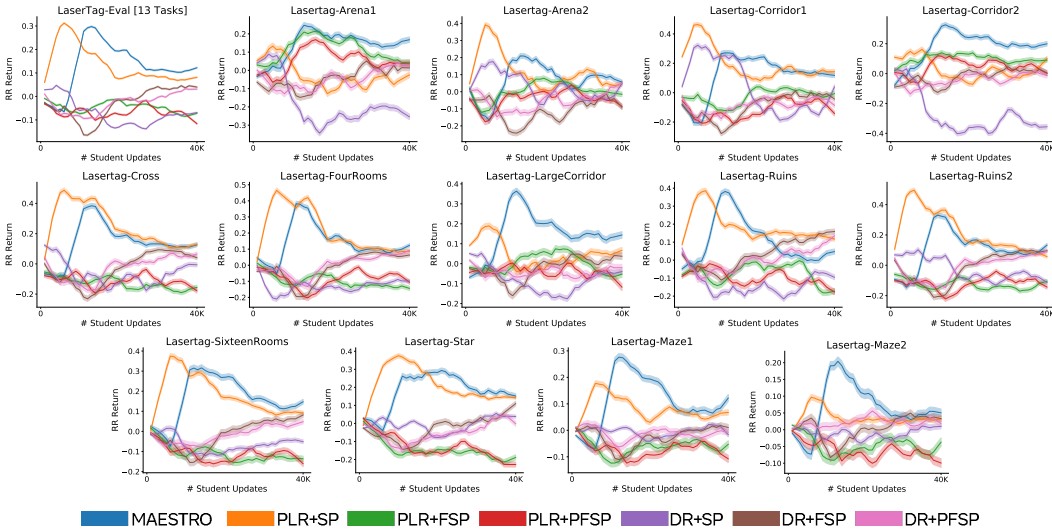

Figure 14: Round-robin return in cross-play between MAESTRO and 6 baselines on all LaserTag evaluation environments throughout training (combined and individual). Plots show the mean and standard error across 10 training seeds.

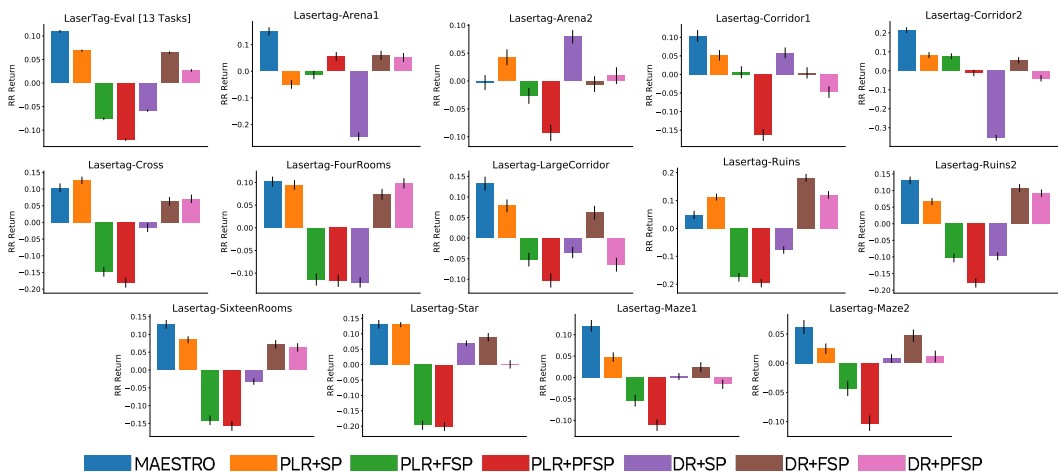

Figure 15: Returns in a round-robin tournament between MAESTRO and 6 baselines on all LaserTag evaluation environments (combined and individual). Plots show the mean and standard error across 10 training seeds.

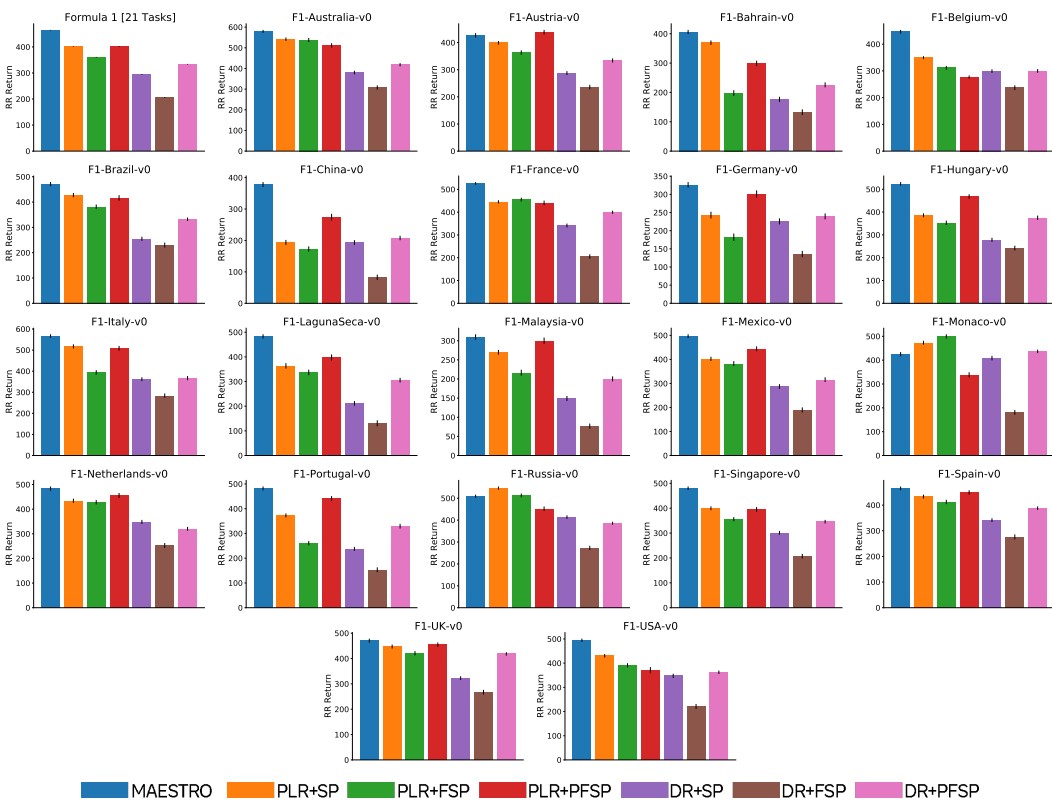

Figure 16: Round-robin returns between MAESTRO and 6 baselines on all Formula 1 tracks (combined and individual). Plots show the mean and standard error across 5 training seeds.

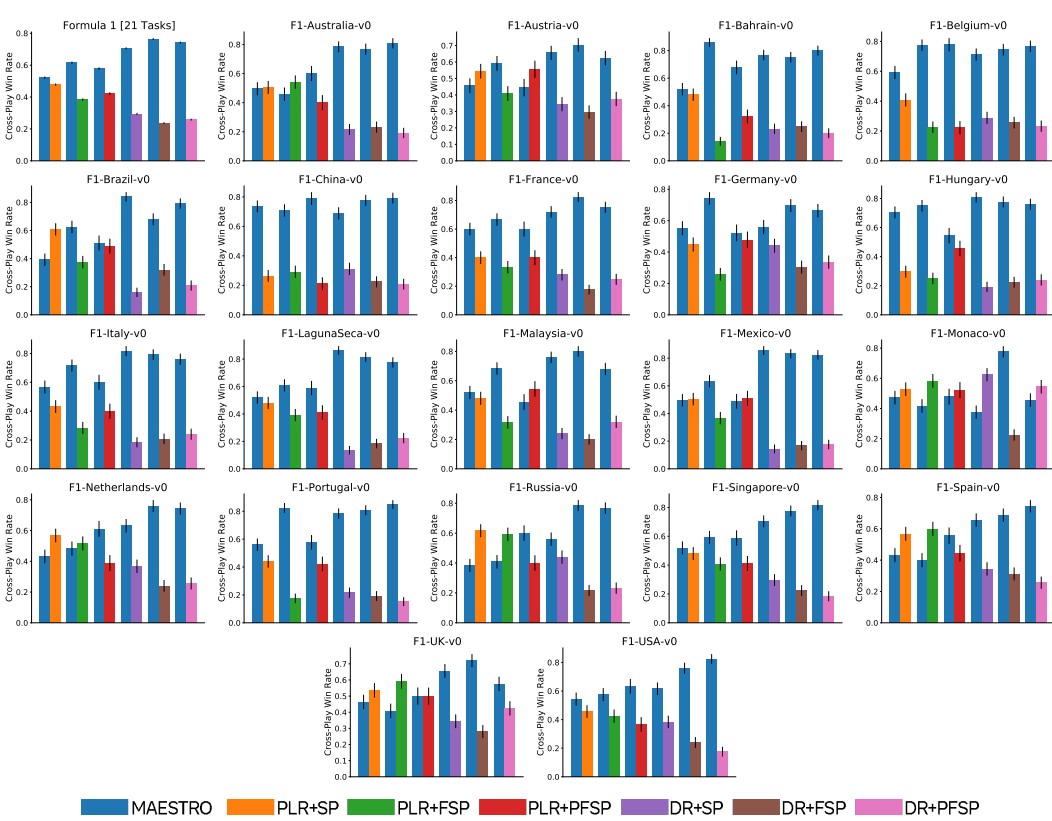

Figure 17: Win rates in cross-play between MAESTRO vs each of the 6 baselines on all Formula 1 tracks (combined and individual). Plots show the mean and standard error across 5 training seeds.

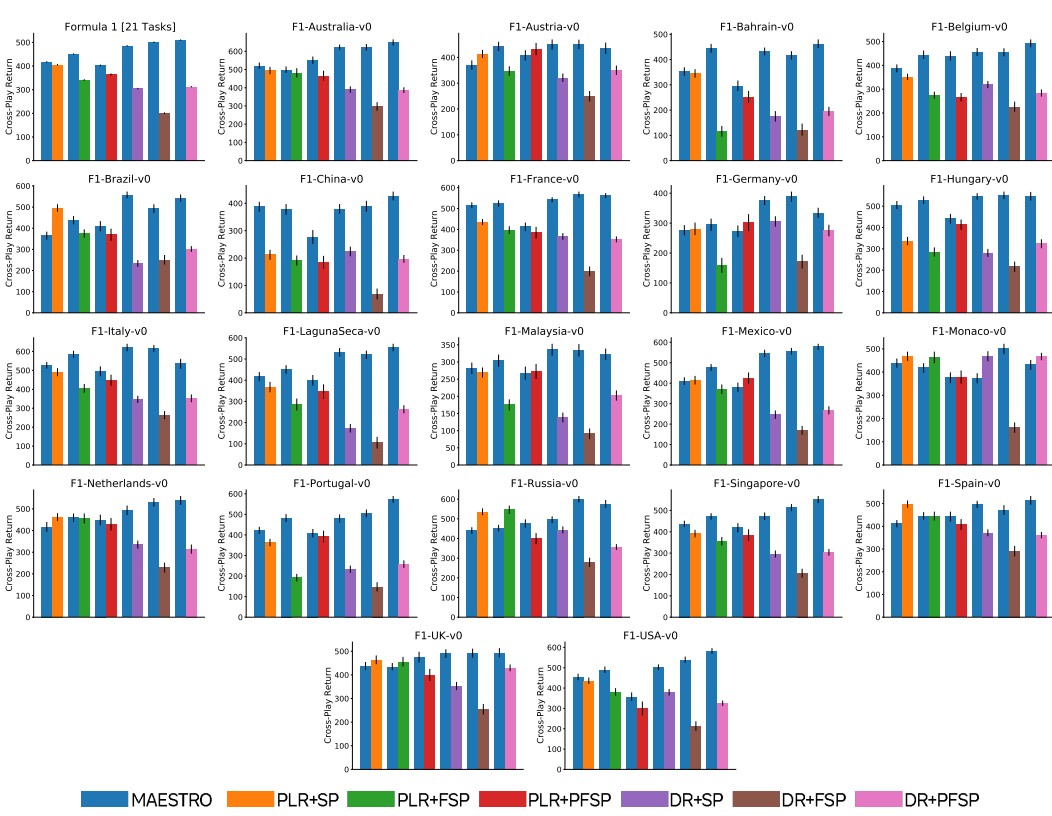

Figure 18: Returns in cross-play between MAESTRO vs each of the 6 baselines on all Formula 1 tracks (combined and individual). Plots show the mean and standard error across 5 training seeds.

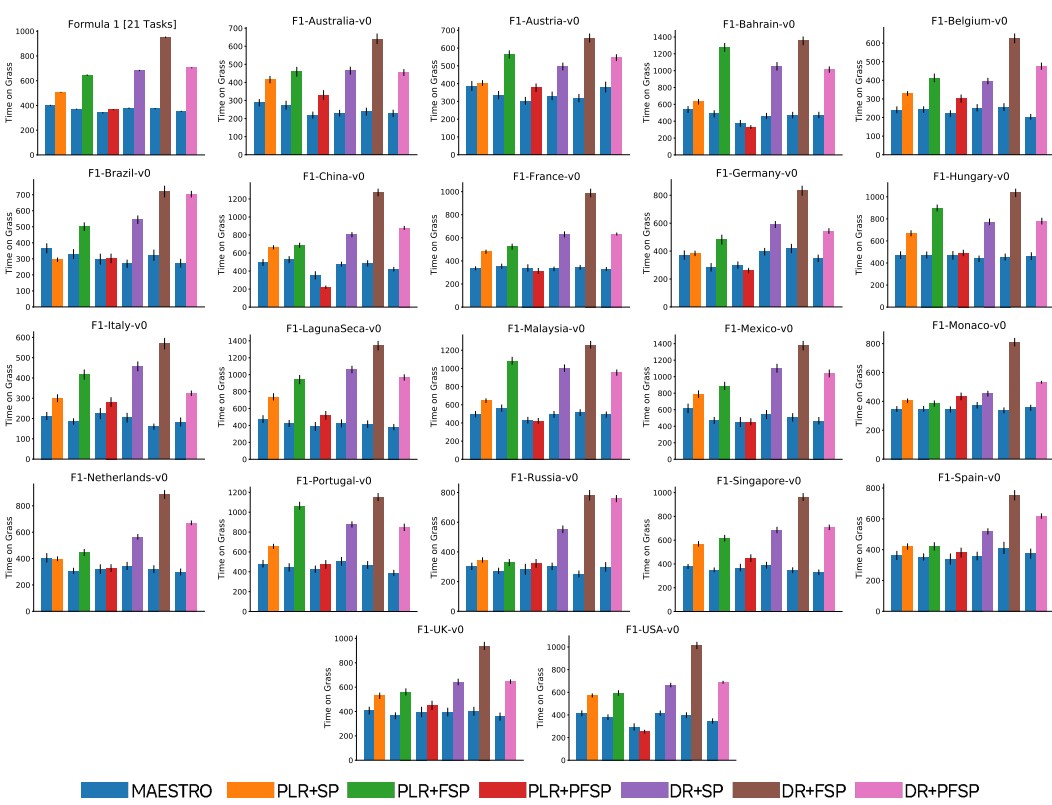

Figure 19: Time on grass during cross-play between MAESTRO vs each of the 6 baselines on all Formula 1 tracks (combined and individual). Plots show the mean and standard error across 5 training seeds.

