# OpenReview forum: "MAESTRO: Open-Ended Environment Design for Multi-Agent Reinforcement Learning"
_ICLR.cc/2023/Conference — ICLR 2023 poster_

### Official Review · Reviewer_AFZu · 2022-10-21

**Confidence:** 3
**Correctness:** 3
**Technical Novelty And Significance:** 3
**Empirical Novelty And Significance:** 2
**Recommendation:** 6

**Clarity, Quality, Novelty And Reproducibility:**

The prose is of average quality and clarity. The approach of considering co-player and game instances together has merit in novelty.

Details for training the policies are not included so the work would not be reproducible without the inclusion of the source code.

**Strength And Weaknesses:**

**Strengths**
 - I have never seen an approach that explicitly accounts for regret with respect to nature and I think that the idea is straightforward and nice.
 - The authors include reasonable ablations of the game instance generation methods as evidence supporting that regret-based prioritization is empirically advantageous.


**Weaknesses**
 - While the idea has a nice intuitive appeal, it's not clear to me that maintaining buffers for the highest-regret game instances is the most sensible. If these settings rarely if ever occur in practice (or under the evaluation distribution) then your learned strategy may compromise its performance in general for these rare cases.
 - Could the authors explain how the proposed theoretical guarantees of Maestro (convergence to BNE) hold when it relies on an expectation over game instances and in practice a highly bias discrete sample of these instances is accounted for? It seems to me that there is an important gap and that the authors should make this clear within the paper.
 - The authors name their algorithm in the style of Policy-Space Response Oracles (PSRO) a general game solving algorithm. Despite this, only PSRO's predecessors are extended (FP, FSP) to adtop their extension when PSRO can naturally be extended in the same way. This is an odd gap that makes me suspicious as a reader. Did the authors try this and it not work?
 - Minimum expect return is an atypical measure of performance for the game solving methods. Could the authors please comment on the regret and the coplayer return as it relates to the relative performance of each method. It seems this measure may lend advantage to the method introduced as it optimizes for the worst-case game instances.


**Summary Of The Paper:**

This paper investigates learning training agents in games where nature samples the games initial state before play. In such a games certain strategies may be at an advantage when the settings play into their strengths. Present methods consider the performance of each strategy with respect to an approximate expectation of the initial state sampling distribution. This work looks at game solving while taking into account the distribution of game settings. Specifically, they propose a population based-training regime where iteratively a policy is optimized to perform well against each coplayer strategy in the worst-case (where buffers of game configurations are used to approximate the game distribution). They demonstrate their approach on a discrete laster-tag gridworld and a continuous racing game.

**Summary Of The Review:**

The paper introduces an intuitively nice idea and offers some reasonable empirical analysis of the method. However, I am not convinced that the theoretical results apply to the method, and am suspicious that about the empirical results and that the algorithm's namesake is only a footnote at the end of the paper.

I think with some minor edits and additional empirical results this paper could be ready for publication.

---

> ### Author Response · Authors · 2022-11-12
> **Response to Reviewer AFZu (Part 2/2)**
>
> ### On the choice of performance metric for cross-play
>
> - The minimum expected return (the worst-case return over co-player policies) can be thought of as giving a better estimate of the **exploitability**, which is a common measure in multi-agent RL research and which MAESTRO optimises against using the minimax-regret objective.
> - We also report the **round-robin (RR) tournament** performances between MAESTRO and all baselines. A round-robin tournament is a competition where each contestant meets every other participant. Full RR returns for LaserTag are shown in Figures 15 and 17 in Appendix E.1, where MAESTRO convincingly outperforms the baselines (Figure 3 (Right) in Section 5 of the main paper also includes RR returns for the final models). Full RR returns for MCR are shown in Figure 18 in Appendix E.3, where MAESTRO again convincingly outperforms the baselines.
>
> ### REFERENCES
>
> 1. Milnor et al, *Games against nature,* RAND PROJECT AIR FORCE SANTA MONICA CA, 1951.
> 2. Peterson. *An introduction to decision theory,* Cambridge University Press, 2017.
> 3. Halpern et al, *Weighted sets of probabilities and minimax weighted expected regret: new approaches for representing uncertainty and making decisions*, UAI 2021
> 4. Dennis et al, *Emergent Complexity and Zero-shot Transfer via Unsupervised Environment Design*, NeurIPS 2022 (Oral).
> 5. Gur et al, *Environment Generation for Zero-Shot Compositional Reinforcement Learning*, NeurIPS 2021.
> 6. Jiang et al, *Replay-Guided Adversarial Environment Design*, NeurIPS 2021.
> 7. Parker-Holder et al, *Evolving Curricula with Regret-Based Environment Design,* ICML 2022.
> 8. Jiang et al, *Grounding Aleatoric Uncertainty in Unsupervised Environment Design*, NeurIPS 2022.

---

> ### Author Response · Authors · 2022-11-12
> **Response to Reviewer AFZu (Part 1/2)**
>
> We are very grateful to Reviewer AFZu for their detailed feedback and suggestions which we used to improve the manuscript. It is nice to hear that the reviewer found the ideas in our method to be both novel and straightforward.
>
> We address your concerns below and hope that this will lead to strengthening your recommendation or letting us know what still stands in the way, so that we may further improve the manuscript.
>
> ### On training the agent on high-regret game instances
>
> This is a very good question and we would like to clarify that:
>
> 1. Since the underlying distribution of environments is assumed to be unknown, any distribution we specify would induce some bias, even the uniform distribution, as human-built levels certainly are not uniformly random. You are thus correct that minimax-regret level prioritisation produces a bias (which is investigated in [8]), but it is crucial to note that this bias is importantly different from a worst-case bias. An environment distribution in which an agent must incur a high regret is one in which there is important unresolved ambiguity. In particular, there must be a lot of available utility, since otherwise there is nothing to regret. This leads to prioritising situations in which the agent could most affect the outcome. Moreover, if the sources of ambiguity are large enough that no policy can achieve low regret on these levels, then there is some important tradeoff that has been left unspecified about the level distribution, in which case no policy could ensure good performance without knowing the underlying distribution of environments.
> 2. For these reasons maximising the regret of the student agent is a widely used and desirable objective both in the decision theory literature on decisions under ignorance [1-3] and in single-agent UED approaches (e.g. see [4-8], all accepted at major venues) that has several advantages. First, minimax-regret UED approaches have theoretical guarantees in equilibria. Second, these methods have demonstrated robust zero-shot generalisation to different game instances. In fully observable settings, they have robustness guarantees over the entire support of environments - in all environments, they perform optimally (in single-agent settings) or reach Nash equilibrium (in two-player zero-sum settings) as we should in our paper.
> 3. While regret-based UED approaches have obvious advantages, we appreciate that identifying the best strategies for prioritising environment/co-player pairs is an interesting research question that future work should address. For example, [Korshunova et al (2021)](https://openreview.net/pdf?id=1-j1aJfCKa9) investigate various metrics (such as return dispersion) for estimating the learning potential of agents in replay-guided methods and conclude that PLR with L1 loss (which does not clearly align with regret) performs better than previously used regret approximations like the trajectory-averaged positive value loss.
>
> ### On the theoretical guarantees of Maestro
>
> - Thank you for raising this subtle but important point. We only provide MAESTRO’s theoretical guarantees at equilibrium. Our results do not provide any characterisation of its convergence to such equilibria. This analysis is similar to that employed on previous single-agent UED techniques [4-8]. We have made it clear in the updated manuscript that convergence guarantees in underspecified environments are an interesting area for future work.
>
> ### On PSRO and its extensions
>
> Thank you for pointing this out. We would like to clarify that:
>
> - We do not only use PSRO’s predecessors (FSP) but also its successor, namely Prioritized Fictitious Self-Play (PFSP). Introduced in [(Vinyals et al, 2019 - Nature)](https://www.nature.com/articles/s41586-019-1724-z), PFSP is a practical variant of PSRO and has been used in AlphaStar for training grandmaster-level agents in StarCraft II.
> - In its standard form, PSRO is a computationally expensive method, as it requires training n policies in n-player games. Applying it to UED, a problem that is already computationally expensive given the large number of diverse environments used for training would render our experiments prohibitively expensive. Therefore, all of our agents and baselines train only a single policy, whilst also using older “fictitious” checkpoints of the student, whereas PSRO would require twice as many training updates. We have clarified this in the updated manuscript while maintaining that concurrent and continued training of several unique policies in underspecified multi-agent problems remains a generally fruitful research direction for future work.

---

> > ### Comment · Reviewer_AFZu · 2022-11-17
> > **Response**
> >
> > Thank you for taking the time to address my comments in great detail.
> >
> > I appreciate the discussion on high-regret game instances and theoretical guarantees, they alleviate my major concerns.
> >
> > However, the commentary about PSRO is unfortunately not completely accurate. PSRO is a class of algorithms for solving games that interleaves empirical game solving and deep reinforcement learning to compute response policies. FSP and PFSP can be cast as instances of PSRO, and are not predecessors/successors necessarily (FSP paper came out before the PSRO paper so likely inspired it, so may be considered a predecessors in some sense). AlphaStar does not simply use PFSP, but a much more complicated League system with 3 different notions of population sampling methods that do not guarantee to maintain the qualities of fiction or self play. PSRO trains n*E policies in n-player games, where E is the number of iterations run.
> >
> > I still like the work, but I remain firm on my position that the naming is exceptionally misleading. I would advocate for distinguishing the work from PSRO through modifying the name, or truly apply it to PSRO.

---

> > > ### Author Response · Authors · 2022-11-17
> > > **We updated the name of our method**
> > >
> > > We thank the reviewer for their response and the useful feedback. It is great to hear that our response has addressed the reviewers major concern regarding high-regret games and theoretical guarantees.
> > >
> > > After careful consideration, we agree with the reviewer that the similarity between our method's name and PSRO might be confusing to the readers. **We therefore have modified the name of our method in the updated manuscript.** Our method is now called *Multi-Agent Environment Design Strategist for Open-Ended Learning* (MAESTRO). We hope that this resolves your remaining concern and that you will consider increasing your rating.
> > >
> > > Once again, we thank the reviewer for their active engagement.

---

> > > ### Author Response · Authors · 2022-12-07
> > > **Addressing your remaining concern - changing the name of the method**
> > >
> > > We wanted to remind the reviewer AFZu that we have updated our manuscript in response to their feedback. We have made several changes, including changing the name of our method to distinguish it from PSRO. We believe that these changes address your remaining concerns and would greatly appreciate if you could consider updating your score and increasing your support for our paper.
> > >
> > > Thank you for taking the time to review our manuscript. We look forward to hearing from you.

---

### Official Review · Reviewer_wyTE · 2022-10-24

**Confidence:** 4
**Correctness:** 4
**Technical Novelty And Significance:** 3
**Empirical Novelty And Significance:** 4
**Recommendation:** 8

**Clarity, Quality, Novelty And Reproducibility:**

* This paper is of high-quality and the exposition is exceptionally clear.

* The method presented appears to be novel, though it primarily serves to combine ideas in prior co-player/environment autocurricula works. The theoretical results and empirical results appear to be novel.

* The paper appears to be reproducible from the details in the paper alone, but the authors also promise to release code along with the camera-ready version.

**Strength And Weaknesses:**

Strengths:

* The paper is exceptionally well-written, well-organized, and clear. Care was clearly taken in the formatting too, and the paper is even aesthetically pleasing.

* The method is well-motivated, and the discussion surrounding related works is comprehensive.

* The method details are described clearly, and useful reference points are provided (e.g. Maestro being a variant of PFSP in fixed singleton environment setting).

* The theoretical results appear to be useful and correct (though I did not read in detail).

* The “illustrative example” is very useful and clearly expresses the motivation.

* The empirical results are compelling and comprehensive, and all of the appropriate baselines (to the best of my knowledge) are included. The results deliver what the method promises, and they also provide more insight into the nature of the problem (via the analysis of the regret landscape) and the nature of the emergent curriculum. The emergent curriculum analysis is very cool (e.g. the wall percentage increasing and grid size decreasing).

Weaknesses:

* None major.

* Typo in middle of 3.2.1 (random section symbol).

**Summary Of The Paper:**

This paper proposes Maestro, an open-ended multi-agent learning method that extends UED to the multi-agent setting, combining the benefits of both environment and co-player autocurricula approaches. The work claims to be the first to adequately address the dependency between co-players and environment design when shaping the autocurricula. The theoretical results presented show that Maestro both produces a regret maximizing distribution over environments and co-players and attains Nash Equilibrium in each supported environment. The authors pit Maestro against a cross-product of relevant environment/co-player autocurricula works in the literature, and demonstrate in two diverse environments (grid-world w/ sparse rewards and continuous control w/ dense rewards) that Maestro performs the best in cross-play. Additional supporting results are presented, including analysis of the regret landscapes (highlighting the importance of a curricula over the joint space), evaluation versus specialist agents (demonstrating Maestro’s robustness), and analysis of the emergent curriculum (highlighting interesting qualitative differences between Maestro’s curricula versus the baselines’).

**Summary Of The Review:**

I vote for this paper’s acceptance. The method is well-motivated, the writing is clear, the results are interesting and compelling, and the findings and resources will likely be of use to the field.

---

> ### Author Response · Authors · 2022-11-12
> **Response to Reviewer wyTE**
>
> We thank the Reviewer wyTE for their high praise of our manuscript. It is great to hear that the reviewer found that our paper is exceptionally well-written and well-motivated, and our results compelling and comprehensive with respect to all of the appropriate baselines. Also, thank you for pointing out the typo which we already fixed.
>
> Please let us know if you have any recommendations for further improving the paper and strengthening your support.

---

### Official Review · Reviewer_GPFe · 2022-11-04

**Confidence:** 3
**Correctness:** 4
**Technical Novelty And Significance:** 2
**Empirical Novelty And Significance:** 2
**Recommendation:** 6

**Clarity, Quality, Novelty And Reproducibility:**

Clarity: Good

Quality: Good

Novelty: This is a straight forward extension to the PLR and Robust PLR (Jiang et al, 2021)

Reproducibilty: Ok

**Strength And Weaknesses:**

Strengths:
- Extending the UED to multi-agent setting is an important problem to tackle; and for the chosen environments, the performance of the proposed algorithm was compared with all the relevant work.

Weaknesses:
- While the authors claim that the proposed framework is applicable to all multi-agent RL settings, the results were shown only on 2-player competitive games. It would add more value to see at least one cooperative setting and one multi-player (more than 2 players) setting.
- No analysis was provided on the memory complexity or time complexity of the proposed algorithm. These are important because the algorithm maintains an independent replay buffer for each player and with increasing number of players, it also becomes unclear of how the algorithm can effectively simultaneously choose the best policy for each player and environment parameters. Without this analysis, it is unclear if the proposed framework and the algorithm can extend beyond the 2-player setting.


**Summary Of The Paper:**

The paper extends the framework of unsupervised environment design to multi-agent setting and demonstrate its effectiveness on two 2-player competitive games.

**Summary Of The Review:**

This paper addresses an important problem of open-ended learning in a multi-agent setting. While the proposed algorithm seems like a straight forward extension of existing algorithms (PAIRED, PLR, Robust PLR), the authors showed a convincing performance of the proposed algorithm in comparison to all the relevant work.

---

> ### Author Response · Authors · 2022-11-12
> **Response to Reviewer GPFe**
>
> We thank the Reviewer GPFe for their feedback that we used to improve our paper. It is great to hear that the reviewer found that MAESTRO addresses an important problem and that the paper showed a convincing performance of the proposed algorithm in comparison to all the relevant work. We address your main concerns below.
>
> ### On the tested multi-agent settings
>
> - We do not claim that our proposed method is applicable to *all* multi-agent RL settings. We made it clear to the reader early on that MAESTRO is designed for 2-player competitive settings and we concentrate our empirical and theoretical analysis on such settings only. That said, the formalism of the UPOSG problem for underspecified multi-agent environments in Section 2 is appropriate for n-player general-sum games. We agree that applying UED in cooperative and n-player settings are exciting research directions and hope that our work will encourage future work in this space.
>
> ### On the space and time complexity
>
> - MAESTRO is a computationally inexpensive method. Given a multi-agent RL framework and a distribution of multi-agent environments, MAESTRO does not meaningfully change the runtime of the framework since it only stores a fixed buffer of checkpoints and efficiently estimates the regret approximations. Space complexity is also insignificant, considering that our models/environment checkpoints only use a small fraction of memory used by modern multi-agent RL frameworks.
> - More specifically, the space complexity of *additional components of MAESTRO* is equal to $m * (b + s)$ where $b$ is the size of per-agent environment buffer, $m = \frac{\text{number of total updates}}{\text{agent checkpoint interval}}$ is the maximum size of co-player population and $s$ is the mode size of a single agent (i.e., weights of the neural network). While MAESTRO’s space complexity is $m$ times larger than that of PLR for the same individual environment buffer size, the best hyperparameters for the PLR-based baselines use a much larger environment buffer than a per-agent buffer used by MAESTRO. For example, our PLR-based baselines use the buffer size of 4000 and 8000 for LaserTag and MultiCarRacing, respectively, whereas MAESTRO’s uses 1000-sized buffers for each member of the population with $m$ having the values of 8 and 16 on those problems.
> - Estimating the regret of each co-player is O(1), given the regret estimates of all environments in the buffer are pre-computed (when the agent last played on that environment). Hence, MAESTRO has the same time complexity *for regret estimation* as Robust PLR with an environment buffer of size $m*b$.
>
> We hope that our clarifications will lead you to consider increasing your support for our paper or explaining what still stands in the way, so that we may further improve it.

---

### Author Response · Authors · 2022-11-17
**Please let us know if we have addressed your concerns**

We once again extend our gratitude to all reviewers for their time, the positive reception of our work, and the useful feedback. We have provided individual responses to all reviewers and updated the manuscript to improve the clarity of the paper, as well as included additional results (as requested by AFZu).

As the author-reviewer discussion phase comes to a close tomorrow, we are wondering if you could please let us know if our rebuttal addresses your remaining questions and concerns, and whether you would consider increasing your rating of our paper in light of our responses. If not, we would appreciate it if you could share what more stands in the way of strengthening your recommendation for our manuscript.

---

### Decision · Program_Chairs · 2023-01-20

**Decision:**

Accept: poster

**Justification For Why Not Higher Score:**

The work is very solid and incremental to extend environment design ideas to two-player competitive games. But this is not a general approach for any multi-agent game hence I lean towards a poster instead of a spotlight.

**Justification For Why Not Lower Score:**

There are no issues in this paper to be rejected.

**Metareview: Summary, Strengths And Weaknesses:**

This paper extends the idea of unsupervised environment design to two-player competitive settings. The authors introduce the MAESTRO algorithm which produces a joint curriculum over environments and players. Authors show that MAESTRO agents are robust when it comes to cross-play.

Reviewers raised several concerns and the authors have addressed them all. Reviewers agree that the paper is well-written. The authors also changed the name of the algorithm to avoid confusion. This is a solid work that is worth publishing at ICLR. I recommend acceptance.

**Note From Pc:**

if the above contains the word "oral" or "spotlight" please see: "oral" presentation means -> notable-top-5% and "spotlight" means -> notable-top-25%. As stated in our emails, we are disassociating presentation type from AC recommendations